# Concerted transcriptional regulation of the morphogenesis of hypothalamic neurons by ONECUT3

Maja Zupančič[1,12], Erik Keimpema [1,12,13] ✉, Evgenii O. Tretiakov [1], Stephanie J. Eder [2,3], Itamar Lev[2], Lukas Englmaier [4], Pradeep Bhandari [5], Simone A. Fietz [6], Wolfgang Härtig[7], Estelle Renaux [8], Andreas Villunger [4,9], Tomas Hökfelt [10], Manuel Zimmer[2,11], Frédéric Clotman [8] & Tibor Harkany [1,10,13] ✉

Acquisition of specialized cellular features is controlled by the ordered expression of transcription factors (TFs) along differentiation trajectories. Here, we find a member of the Onecut TF family, ONECUT3, expressed in postmitotic neurons that leave their *Ascl1*[+]/*Onecut1/2*[+] proliferative domain in the vertebrate hypothalamus to instruct neuronal differentiation. We combined single-cell RNA-seq and gain-of-function experiments for gene network reconstruction to show that ONECUT3 affects the polarization and morphogenesis of both hypothalamic GABA-derived dopamine and thyrotropin-releasing hormone (TRH)[+] glutamate neurons through neuron navigator-2 (*NAV2*). In vivo, siRNA-mediated knockdown of ONECUT3 in neonatal mice reduced NAV2 mRNA, as well as neurite complexity in Onecut3-containing neurons, while genetic deletion of *Onecut3/ceh-48* in *C. elegans* impaired neurocircuit wiring, and sensory discrimination-based behaviors. Thus, ONECUT3, conserved across neuronal subtypes and many species, underpins the polarization and morphological plasticity of phenotypically distinct neurons that descend from a common pool of *Ascl1*[+] progenitors in the hypothalamus.

As the neuroendocrine center of the brain, the mammalian hypothalamus contains a manifold of magnocellular and parvocellular neuroendocrine command neurons. These cells are synaptically wired with both local-circuit interneurons and projection neurons for their integration into sensory and executive neurocircuits to precisely define body-wide paracrine signaling through many hypothalamic hormones[1]. The cellular heterogeneity that places the hypothalamus apart from other brain areas (noting that often as few as 500–1000 neuroendocrine cells can control hormonal axes in vertebrates) is determined by a three-dimensional matrix of progenitors distributed

[1]Department of Molecular Neurosciences, Center for Brain Research, Medical University of Vienna, Vienna, Austria. [2]Department of Neuroscience and Developmental Biology, Vienna Biocenter (VBC), University of Vienna, Vienna, Austria. [3]Vienna Biocenter PhD Program, Doctoral School of the University of Vienna and Medical University of Vienna, Vienna, Austria. [4]CeMM Research Center for Molecular Medicine of the Austrian Academy of Sciences, Vienna, Austria. [5]Institute of Science and Technology Austria, Klosterneuburg, Austria. [6]Institute of Veterinary Anatomy, Histology and Embryology, University of Leipzig, Leipzig, Germany. [7]Paul Flechsig Institute for Brain Research, University of Leipzig, Leipzig, Germany. [8]Animal Molecular and Cellular Biology, Louvain Institute of Biomolecular Science and Technology, Université catholique de Louvain, Louvain-la-Neuve, Belgium. [9]Institute for Developmental Immunology, Biocenter, Medical University of Innsbruck, Innsbruck, Austria. [10]Department of Neuroscience, Biomedicum 7D, Karolinska Institutet, Solna, Sweden. [11]Research Institute of Molecular Pathology (IMP), Vienna Biocenter (VBC), Vienna, Austria. [12]These authors contributed equally: Maja Zupančič, Erik Keimpema. [13]These authors jointly supervised this work: Erik Keimpema, Tibor Harkany. ✉e-mail: Erik.Keimpema@meduniwien.ac.at; Tibor.Harkany@meduniwien.ac.at

along both the anterior–posterior and dorsal–ventral axes of the third ventricle, as well as in the preoptic area[2]. Alike their counterparts in other brain areas, hypothalamic neurons use cell-autonomous transcriptional[3] and epigenetic programs[4] in conjunction with activity-dependent intercellular signals, including neurotransmitters[5], to migrate and differentiate into specific neuronal subtypes. It is becoming established that hypothalamic neurons are produced according to the 'cascade diversification model' of neurogenesis, wherein neural progenitors produce intermediate precursors (IPCs) that express either *Ascl1* or *Ngn2* transcription factors (TFs) for pro-neuronal differentiation[6]. Whereas both *Ascl1*+ and *Ngn2*+ IPCs produce glutamatergic neurons[7,8], *Ascl1*+ IPCs are seen as the exclusive source of GABAergic and/or dopaminergic neurons[9]. Accordingly, cascading TFs placed downstream from *Ascl1* can contribute to the migration and particularly differentiation of both glutamate and GABA progenies[3]. However, whether any such TF retains functional competence in phenotypically differentiated progeny, and if so without preference for neurotransmitter identity, remains poorly understood.

One such example is the family of ONECUT TFs, whose evolutionarily conserved members include ONECUT1 (hepatocyte nuclear factor-6; HNF-6), ONECUT2, and ONECUT3, which all contain a bipartite DNA-binding motif consisting of a single CUT domain[10,11] and a distinct homeodomain[12]. ONECUT paralogs regulate cellular fate decisions at the periphery through chromatin remodeling[13], allowing widespread changes in the transcriptional landscape and resulting in the differentiation of hepatocytes[14,15], as well as exocrine acinar and duct cells in the pancreas[16,17]. A role for ONECUT TFs in hypothalamus development was suggested when genome-wide association studies linked deregulated ONECUT3 function to metabolic illness and/or neurological complications[3], even if no individual molecular effector was *de facto* upheld for causality. In contrast, the only data about ONECUT3, the effector TF of the cascade, is its retention by some hypothalamic neurons postnatally[18]. Notably, these include both dopamine/GABA neurons of the periventricular nucleus (PeVN), as well as *Trh*+/glutamate neurons of the lateral hypothalamus (LH). Thus, ONECUT3 could be considered a master gene of a *synapomere*, a stable gene regulatory complex, for concerted transcriptional evolution and development[19] if it subserves a shared role during the lifespan of these phenotypically distinct neuronal subtypes. Yet, ONECUT3 functions outside the proliferative ventricular niche remain unknown.

Here, we show that ONECUT3 is invariably expressed in both GABA/tyrosine hydroxylase (TH)+ and glutamate/thyrotropin-releasing hormone (TRH)+ neurons in the hypothalamus of evolutionarily segregated vertebrate species. Both cell populations arise from a pool of *Ascl1*+ progenitors, with their IPCs residing in the wall of the 3rd ventricle. *Onecut3* is exclusively expressed in neuroblasts exiting the progenitor zone, and remains 'on' throughout cell migration and differentiation in both GABA/tyrosine hydroxylase (TH)+ and glutamate/TRH+ neuroblasts destined to the PeVN and LH/tuberal nucleus (TU), respectively. RNA-seq-based target screens after in vitro gain-of-function identified neuron navigator-2 (Nav2), an F-actin interacting protein[20–22], as a preferred downstream target, possibly linking ONECUT3 to cell-autonomous neuritogenesis in both progenies. Indeed, siRNA-mediated *Onecut3* knock-down in neonatal mice significantly downregulated *Nav2* mRNA, and reduced neurite complexity. Likewise, genetic ablation of *ceh-48*, the *Onecut* ortholog in *C. elegans*, decreased *unc-53* (*Nav2* ortholog) expression, impaired sensory dendritogenesis in amphid neurons, and consequently disrupted sensory transduction in a chemotaxis assay. These data, in conjunction with the pharmacological sensitivity of ONECUT3 action to inhibiting both guanine-nucleotide exchange factor Trio (TRIO) and Ras Homolog Family Member A (RhoA), suggest that ONECUT3 regulates *Nav2* for neurite outgrowth. Thus, we define an ASCL1 → ONECUT3 → NAV2 → RhoA pathway for neuronal differentiation, which operates equally in even far-placed neuroendocrine neurons in the hypothalamus.

## Results

### ONECUT3 expression in the fetal mouse hypothalamus

We first sought to precisely address the expressional onset of ONECUT3 protein and map its progression during hypothalamus development. At embryonic day (E)8.5, ONECUT3 protein could not be detected in the neural tube ectoderm (Fig. 1a). Whole mount histochemistry revealed the first ONECUT3+ structures in the nervous system between E9.5 and E10.5 (Fig. 1b, $c_1$), concentrating at the ventral diencephalon (the prospective hypothalamus), mesencephalon, rhombencephalon, and spinal cord. All ONECUT3+ cells were positioned in a pearl-lace-like configuration at the outer border of the SOX2+ ventral germination (proliferative) zone (Fig. 1d) with ONECUT3+ cells being unequivocally negative for SOX2, a TF enriched in neural progenitors to promote their proliferation[23]. These data suggest that ONECUT3+ cells are an early-born progeny in the prospective hypothalamus. To support this notion, we used a mCherry reporter plasmid injected into the 3rd ventricle for in-utero electroporation at E13.5 (Supplementary Fig. 1a, b). None of the ONECUT3+ neurons (which expressed NeuN in adulthood, Supplementary Fig. $3b_2$) were labeled by mCherry, suggesting that their birth was indeed at an earlier time point.

Considering the nuclear localization of ONECUT3 in SOX2− postmitotic progeny (see above), we took advantage of *Onecut3*-mCherry reporter mice, as well as *Onecut3*-iCre::Ai14 tdTomato and *Onecut3*-iCre::Tau-mGFP[24] compound lines to map the spatial distribution of mCherry+ neurons, and to resolve their axonal projections (mGFP+) over time. Both iCre lines, irrespective of expressing either membrane-bound mGFP or cytoplasmic mCherry, had transgene-labeled progeny in the diencephalon at E9.5 (Fig. 1e, $e_1$). From E10.5, mCherry accumulated in both somata and processes (Fig. 1f–h), with mCherry+ processes coursing towards the midbrain (mesencephalon), hindbrain (rhombencephalon), and spinal cord. We considered these structures as axons because of their co-localization with growth-associated protein-43[25] (GAP43; Fig. $1f_1$). At E11.5, ONECUT3+ cells with spindle-shaped somatic morphologies and long leading and trailing processes were reminiscent of postmitotic neuroblasts engaged in long-range migration (Fig. $1h_1$)[26]. Finally, we imaged intact *Onecut3*-mCherry brains at E14.5 to show that ONECUT3+ territories formed an intertwined chain, stretching primarily along the midline of the early hypothalamus and into the septal area of the fetal forebrain (Fig. 1i). This expression pattern remained unchanged throughout intrauterine development and in adulthood, with a considerable decrease in hypothalamic *Onecut3* mRNA, which we interpreted as the relative enrichment of the hypothalamus of late-born neurons belonging to other subclasses (Fig. 1j and Supplementary Fig. 2).

### ONECUT3 expression in the postnatal mouse hypothalamus

In the immature hypothalamus at E14.5, *Onecut3* mRNA and protein accumulated in the PeVN and LH (Supplementary Figs. 2a, b and 3b–$b_2$), and remained so into adulthood. In contrast, *Onecut1* and *Onecut2* TFs were developmentally restricted as they disappeared in juvenile mice (Supplementary Fig. 2c–$f_1$), recapitulating earlier data on spinal motor neurons[27]. Histochemical mapping of ONECUT3 distribution at postnatal day (P)32 defined a central-to-lateral ribbon-like band of immunoreactive neurons, which stretched from the PeVN through an elongated and narrow corridor (here termed the, MZ) towards the LH/TU, while leaving the paraventricular, dorsomedial (DMH), as well as suprachiasmatic, ventromedial (VMH), and arcuate nuclei (dorsal and ventral boundaries) essentially free of immunoreactivity (Supplementary Figs. 3b, $b_1$ and 4a–$b_2$). These data suggest that hypothalamic ONECUT3 expression marks postmitotic neurons poised to undergo laterally oriented chain migration to populate distant-placed hypothalamic areas.

*E8.5 whole mount*

*E9.5 whole mount ONECUT3*

*ONECUT3 mapping at E10.5*

*E10.5 OC3 expression outside the VZ*

*OC3-iCre::Tau-mGFP*

*OC3-iCre::Ai14*

*OC3-iCre::Tau-mGFP*

*E10.5-E11 OC3-iCre::Tau-mGFP, Onecut3, GAP43*

*OC3-iCre::Ai14*

*E11.5 OC3-mCherry*

*IHC mCherry, OC3*

*E14.5 whole mount brain Onecut3-mCherry*

*qPCR*

**Evolutionarily conserved expression of ONECUT3 in mammals**

The hypothalamus executes essentially the same neuroendocrine functions in all vertebrates[1]. While data exist on ONECUT3 being expressed in neurons (NeuN+/GFAP−) of the adult mouse hypothalamus (see ref. 18. and Supplementary Fig. 3a–b$_2$), here we asked if other phylogenetically segregated adult mammals could also retain ONECUT3 in hypothalamic neurons. Indeed, ONECUT3+ neurons concentrated in the PeVN and LH of subterranean-dwelling naked mole rats (*Heterocephalus glaber*, *Rodentia*; at 3 months and 20 years of age comparable to neonatal and adult mice[28]), with a subset of ONECUT3+ neurons co-expressing TH (Supplementary Fig. 3c, c$_1$). Likewise, the hypothalami of Seba's short-tailed fruit bat (*Carollia perspicillata*) and the Indian flying fox (*Pteropus medius*, both *Chiroptera*), species unrelated to rodents, presented ONECUT3+ foci in the PeVN and the

**Fig. 1 | ONECUT3 expression in the developing mouse embryo. a** Whole-mount immunohistochemistry at E8.5 suggested the lack of ONECUT3 expression. **b** ONECUT3$^+$ neurons were sparsely detected at E9.5 in the spinal cord (arrowheads) with limited presence in the diencephalon (open arrowheads). **c–c$_1$** Whole-mount immunolabelling at E10.5 revealed ONECUT3 protein expression (arrowheads) throughout the central nervous system. **d** ONECUT3 was localized outside the hypothalamic proliferative zone (that is SOX2$^+$). Open arrowheads mark ONECUT3$^+$/SOX2$^-$ cells. **e, e$_1$** ONECUT3$^+$ neuronal processes (arrowheads) at E9.5 were seen in Tau-mGFP (**e**) and tdTomato reporter mice (**e$_1$**). **f, g** The processes (arrowheads) of ONECUT3$^+$ neurons, likely axons, coursed throughout the entire central nervous system by E10.5, as visualized by transgene constructs. Projections (GAP43$^+$) in the spinal cord of the whole mount embryo (**f, f$_1$**) and within the mesencephalon (**g**) were noted. Asterisks label the ONECUT3$^+$ region. **h** In *Onecut3*-

mCherry mice, mCherry exhibited a near-complete overlap with ONECUT3 protein (in green). **i** Whole-mount 3D imaging of *Onecut3*-mCherry brains at E14.5 revealed the co-localization of mCherry with Onecut3 protein throughout the forebrain, mid-, and hindbrain. **j** Expression of *Onecut3* in the hypothalamus decreased with age, as was determined by qPCR and plotted as means ± s.e.m. (*n* = 2/group). For anatomical analysis, immunohistochemistry was performed >3 per developmental age. Source data are provided as a Source Data file. 3V third ventricle, CTX cortex, den diencephalon, E embryonic day, HB hindbrain, HNe hypothalamic neuroepithelium, HYP hypothalamus, IHC immunohistochemistry, LV lateral ventricle, MB midbrain, mes mesencephalon, NT neural tube, OC3 ONECUT3, pros prosencephalon, rhom rhombencephalon, SEP septum, sp spinal cord, tel telencephalon, VZ ventricular zone. Scale bars = 100 μm (**a–c, e, f, h, i**), 50 μm (**d, e$_1$, f$_1$, g**), 20 μm (**h$_1$**).

LH/TU, including both TH and ONECUT3 co-localization (Supplementary Fig. 3d, e). These data suggest that ONECUT3 expression persists into the adulthood of various mammals.

Next, we asked if fetal ONECUT3 expression patterns in these and other mammalian species are near-identical. Undeniably, the embryonic PeVN from Seba's short-tailed fruit bat (Supplementary Fig. 3f), wild boar (*Sus scrofa*; Supplementary Fig. 3g), and sheep (*Ovis aries*, both Artiodactyla; Supplementary Fig. 3h) contained ONECUT3$^+$ neurons (co-labeled with NeuN), many being TH$^+$. The PeVN and LH of embryonic mouse lemurs (*Microcebus murinus*; Supplementary Fig. 3i), a small nocturnal primate, as well as fetal human hypothalami (21 weeks of pregnancy; Supplementary Fig. 3j, j$_1$) also contained ONECUT3$^+$ neurons. These data suggest the evolutionary conservation of ONECUT3 expression across developmental stages and species. Moreover, the incomplete co-localization of ONECUT3 and TH, along with the presence of ONECUT3$^+$ neurons in the LH that seem positionally unrelated to catecholamine cell groups, suggests the existence of hitherto uncharacterized ONECUT3$^+$ neuronal subtypes.

## *Onecut3* marks *Ascl1*$^+$ progenitor-derived GABA and glutamate neurons

The positional segregation of *Onecut3*$^+$ neurons could coincide with them becoming phenotypically distinct[18,29]. This hypothesis can be supported by the ability of *Ascl1*$^+$ progenitors to give rise to both GABA and glutamate neurons in the hypothalamus[7,9]. To test this possibility, we used an open-label single-cell RNA-seq database on developing neurons[3] to show that *Onecut3* is indeed co-expressed with *Ascl1* (Fig. 2a). Next, we performed in situ hybridization to confirm the presence of residual *Ascl1* in *Onecut3*$^+$ neurons at E11.5 (Fig. 2b). We found a cellular transition zone along the 3rd ventricle with antiparallel gradients of *Ascl1* and *Onecut3* expression, indicating the gradual loss of *Ascl1* for *Onecut3*$^+$ neuroblasts to exit the hypothalamic proliferative zone (Fig. 2b$_1$).

Next, we used single-cell RNA-seq data to evaluate the neurochemical identity of hypothalamic *Onecut3*$^+$ neurons in mice. We first found a population of GABA/dopamine neurons containing *Th* and the dopamine transporter *Slc6a3*[30], and co-expressing *Slc32a1* (vesicular GABA transporter) at relatively moderate levels (Fig. 2c, c$_1$ and Supplementary Fig. 5). Considering the protracted developmental trajectory, likely reminiscent of 'reserve pools' of GABA (and GABA-derived dopamine) progeny during hypothalamus development[3], we did not follow neuropeptide-based sub clustering along developmental trajectories. Additionally, we identified a second *Onecut3*$^+$ population co-expressing *Slc17a6* (vesicular glutamate transporter 2) and *Trh* (Fig. 2c, c$_1$ and Supplementary Fig. 5), which showed complete phenotypic segregation in *UpSet* plots (Fig. 2c). Thus, we find a uniform *Ascl1*$^+$ neural progenitor pool to give rise to two distinct populations of *Onecut3*$^+$ neurons (Fig. 2a, c$_1$), compatible with the 'cascade diversification model' of hypothalamic neurogenesis[7]. We then resorted to in situ hybridization and immunohistochemistry in reporter mice to validate the single-cell RNA-seq data. Firstly, we overlaid both *Trh* and

*Th* on *Onecut3* mRNA from E14.5 into adulthood. At E14.5, *Onecut3*$^+$/*Th*$^+$ neurons exclusively populated the PeVN along its rostrocaudal axis (Fig. 3a–a$_2$ and Supplementary Fig. 4a–a$_2$)[3]. In contrast, *Onecut3*$^+$/*Trh*$^+$ neurons showed zonation from the medial and lateral POA along the retrochiasmatic nucleus and the LH/TU (Fig. 3a–a$_2$). We found an equivalent configuration postnatally (Fig. 3b–d$_2$ and Supplementary Fig. 4b–b$_2$), suggesting that these hypothalamic structures were indeed patterned before birth. Moreover, we localized *Onecut3*$^+$/*Trh*$^+$ neurons in the mediolateral zone (MZ), positioned between the dorsomedial and ventromedial hypothalamic nuclei (DMH/VMH) (Fig. 3b–d$_2$), resembling a 'bridge' between the 3rd ventricle and the LH.

RNA distribution was validated in reporter mice in combination with immunohistochemistry. Both GAD65-GFP (Fig. 4a) and GAD67-GFP mice (Fig. 4a$_1$) had neuroblasts that adopted a GABA phenotype in the hypothalamic transition zone at E12.5[31,32]. Many of these GABA neurons were labeled for TH (Fig. 4a, a$_1$). GABA/TH$^+$/ONECUT3$^+$ neurons exclusively populated the PeVN by birth (Fig. 4b) with some retaining GAD65, but not GAD67, postnatally (Fig. 4c, d). In contrast, ONECUT3$^+$ neurons facing the pial surface were devoid of GFP signal in either reporter line (Fig. 4b$_1$). In the territory corresponding to the *Slc17a6*$^+$ pre-LH area[33], we detected co-labeling between *Slc17a6*, *Trh*, and *Onecut3* mRNA after E11.5 (Fig. 4e) and into adulthood, as visualized by pro-TRH immunohistochemistry (Fig. 4f–g$_1$). Finally, we could reveal that GAD65$^+$/ONECUT3$^+$ neurons were bipolar, whereas TRH$^+$/ONECUT3$^+$ cells were multipolar (Fig. 4h)[34]. These data suggest the existence of morphologically distinct populations of ONECUT3$^+$ GABA/dopamine and glutamate/TRH neurons that reside in separate anatomical regions, positioned amongst neurons matching their own neurotransmitter identities (Fig. 4i and Supplementary Fig. 6a, b).

## ONECUT3 overexpression induces neuron-like differentiation

As ONECUT3 is expressed in subpopulations of both GABA and glutamate neurons, we hypothesized that ONECUT3 would not drive the acquisition of neuronal identity alone, but rather underpin a concerted differentiation/maturation program by a shared regulon[3,19]. Firstly, we screened for molecular targets downstream from ONECUT3 in relation to morphogenesis in Neuro-2a cells with a gain-of-function approach. Neuro-2a cells were chosen because they satisfied the following criteria: i) they respond to morphogens by neurite extension[35–37]; ii) *all-trans* retinoic acid induces many pro-neuronal genes that characterize ONECUT3$^+$ neurons, particularly *Th*, *Trh*, *Slc17a6*, and *Gad1/2*[38,39], and iii) Neuro-2a cells do not endogenously contain ONECUT3 protein, offering unbiased overexpression. When transiently transfecting Neuro-2a cells with an *Onecut3*-containing plasmid, their proliferation was reduced as revealed by the lack of phospho-histone H3 (pHH3), a mitotic marker, in transfected cells (93.59% of Onecut3$^+$ cells lacking pHH3; *p* < 0.001; *n* = 259 cells; Fig. 5a, a$_1$, d$_1$). ONECUT3 did not co-localize with cleaved caspase-3, a pro-apoptotic protein, either, suggesting the lack of arbitrary cell death due to overexpression (Fig. 5b). Instead, Neuro-2a cells increased their somatic diameter (control:18.0 μm vs transfected: 24.2 μm; *p* < 0.001; *n* = 47

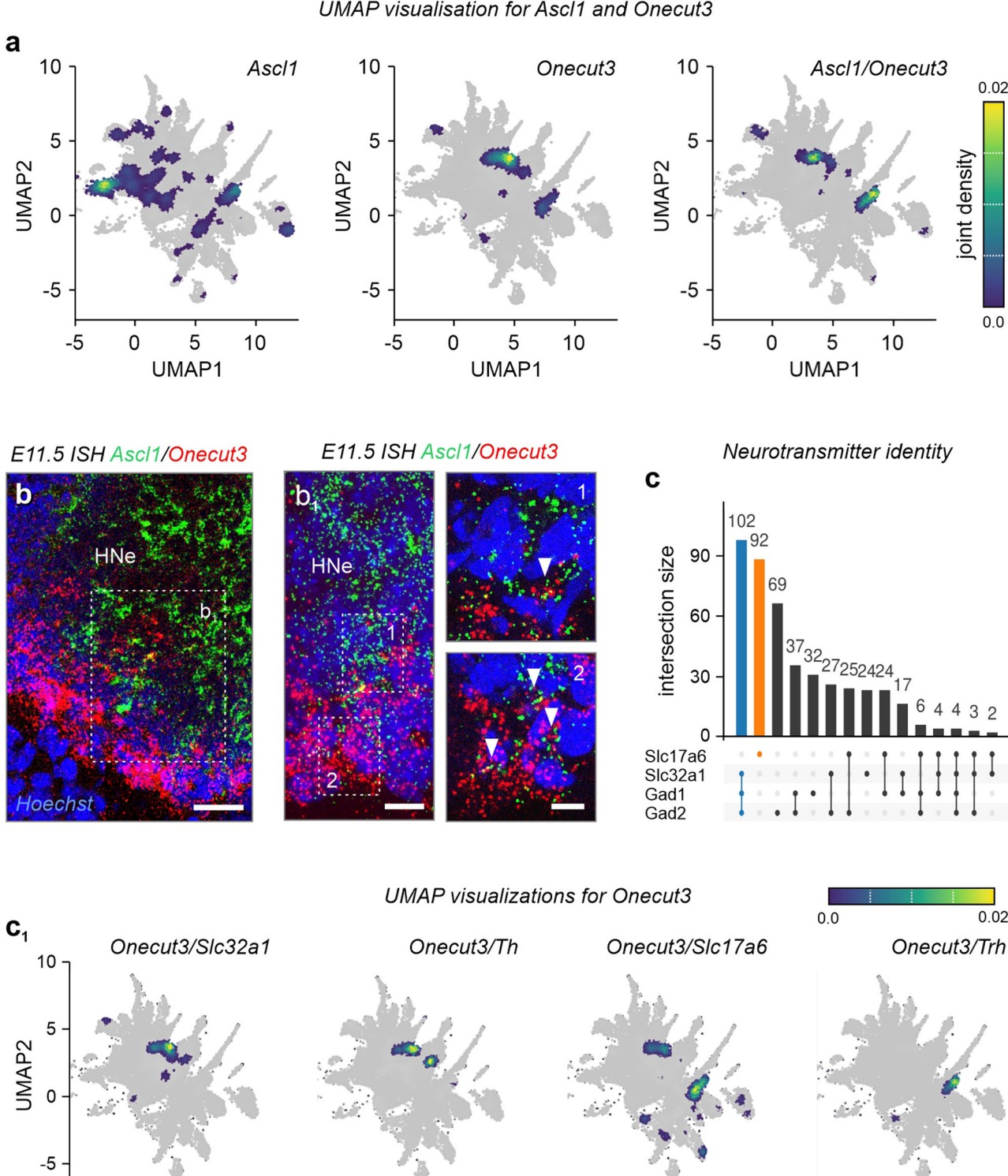

**Fig. 2 | Neurochemical heterogeneity of ONECUT3⁺ neurons within the hypothalamus.** **a, b₁** *Onecut3*⁺ hypothalamic neurons belong to the *Ascl1* progeny, as supported by both UMAP visualization of single-cell RNA-seq data (scRNA-seq; **a**) and in situ hybridization on E11.5 (**b, b₁**). In situ hybridizations were repeated at least two times. **c, c₁** When probed for neurotransmitter and neuropeptide identity in an open-label scRNA-seq dataset combined from E15.5-P23[3], *Onecut3*⁺ neurons formed two distinct groups marked by either *Slc32a1/Th* or *Slc17a6/Trh* expression. UpSet plot (**c**) depicts co-expression frequency amongst *Onecut3*⁺ neurons with *Slc1716*, *Slc32a1, Gad1*, and *Gad2*. *Onecut3*-expressing *Slc32a1/Th*⁺ and *Slc17a6*/Trh⁺ populations were highlighted on UMAP plots (**c₁**). E embryonic day, HNe hypothalamic neuroepithelium, *Th* tyrosine hydroxylase, *Trh* thyrotropin-releasing hormone, UMAP uniform manifold approximation, and projection. Scale bars = 100 μm (**b**), 50 μm (**b₁**), 20 μm (1, 2).

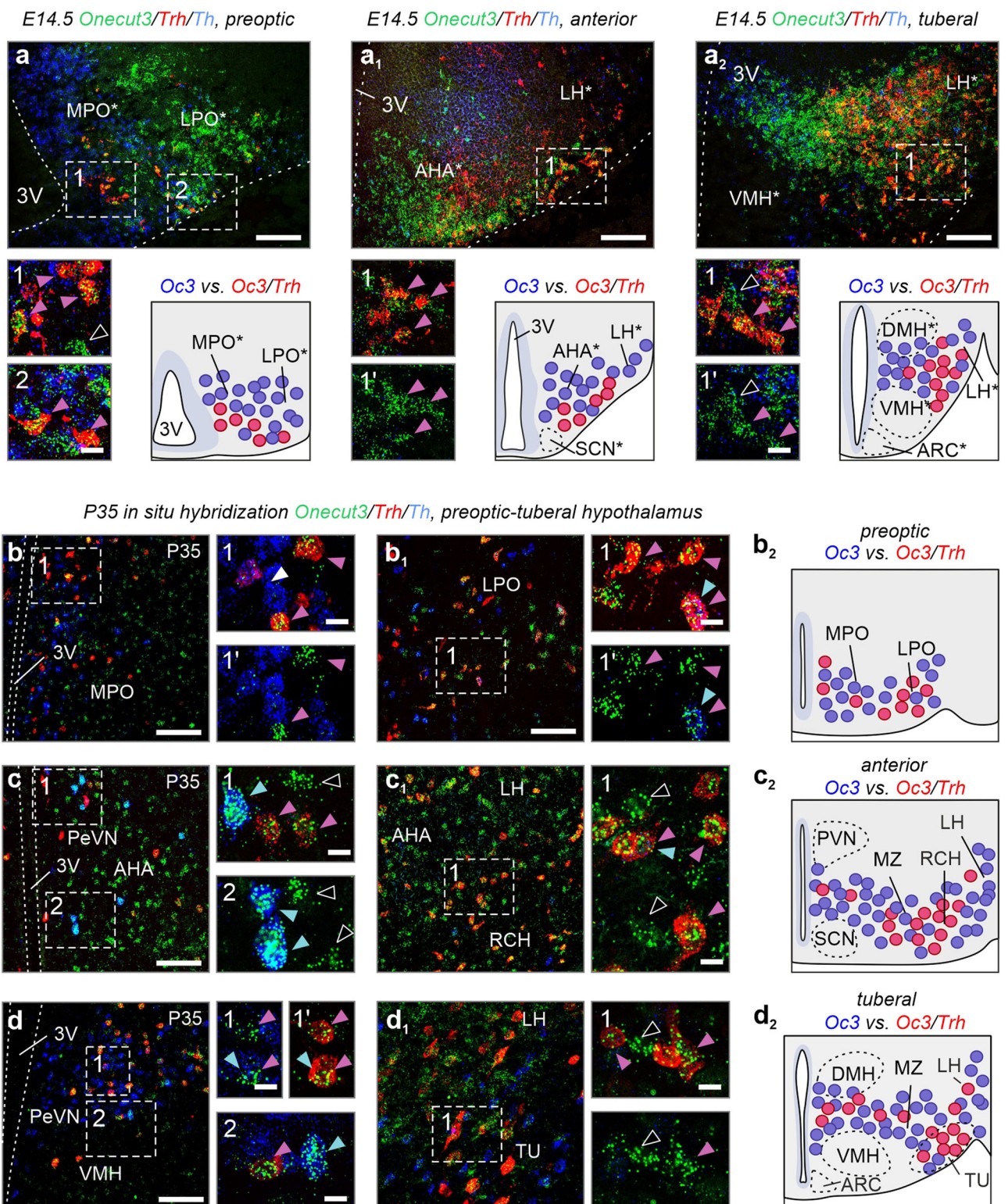

**Fig. 3 | *Trh* and *Th* in *Onecut3*+ hypothalamic neurons. a–d₂** Hypothalamic expression of *Onecut3* and *Trh* and/or *Th* across different developmental stages, as shown by in situ hybridization. At E14.5, *Onecut3*+/*Trh*+ double-labeled neurons were found in the preoptic (**a**), anterior (**a₁**), and tuberal (**a₂**) hypothalamus. Note the increasing density of *Onecut3*+/*Trh*+ neurons in the lateral direction (**a₁, a₂**). A similar pattern was maintained in the adult preoptic (**b–b₂**), anterior (**c–c₂**), and tuberal (**d–d₂**) hypothalamus. *Onecut3*+/*Trh*+ neurons are concentrated in the lateral preoptic area (**b₁**), retrochiasmatic (**c₁**), and tuberal nuclei (**d₂**). *Onecut3*+/*Th*+ neurons were restricted to the PeVN. Pink arrowheads mark *Onecut3*+/*Trh*+ neurons, whereas blue arrowheads point to *Onecut3*+/*Th*+ neurons. Asterisks mark prospective hypothalamic regions. In situ hybridizations were repeated at least two times. 3V third ventricle, AHA anterior hypothalamic area, DMH dorsomedial hypothalamus, E embryonic day, LH lateral hypothalamus, LPO lateral preoptic area, MPO medial preoptic area, MZ mediolateral zone, P postnatal day, PeVN periventricular nucleus, RCH retrochiasmatic nucleus, TU tuberal nucleus, VMH ventromedial hypothalamus, scale bars = 100 μm (**a–d₁**), 20 μm (insets).

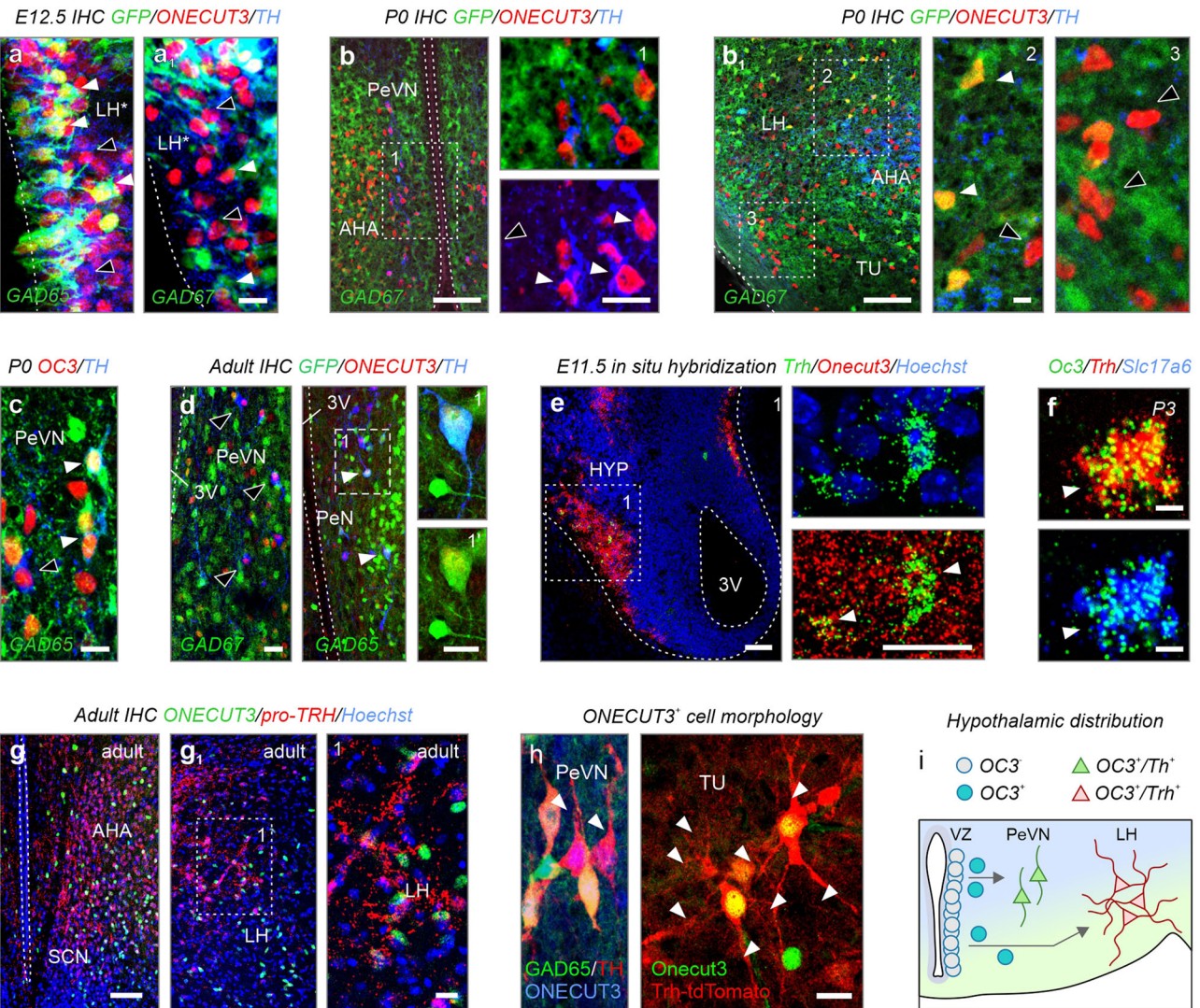

**Fig. 4 | Neurotransmitter identity of ONECUT3⁺ neurons. a, a₁** Immunohistochemistry in GAD65-GFP and GAD67-GFP mice at E12.5 demonstrated that a subpopulation of ONECUT3⁺ neurons co-expressed *Gad1/Gad2* in the hypothalamic neuroepithelium (arrowheads). Open arrowheads point to GFP⁻/ONECUT3⁺ cells. **b, b₁** Immunohistochemistry in GAD67-GFP mice at P0. ONECUT3⁺/GFP⁺ neurons were observed within both anterior (**b**) and lateral (**b₁**) areas (arrowheads). Note the absence of GFP in ONECUT3⁺/TH⁺ periventricular neurons (open arrowheads). **c** ONECUT3⁺/TH⁺ neurons expressed GAD65 in the PeVN (arrowheads) at birth. **d** Immunohistochemistry in adult GAD67-GFP (left) and GAD65-GFP (right) transgenic mice. *Onecut3⁺*/TH⁺ neurons were GFP⁺ in GAD65-GFP (arrowheads) but not in GAD67-GFP mice (open arrowheads). **e, f** Early embryonic (E11.5) and postnatal in situ hybridization revealed the presence of *Onecut3⁺/Trh⁺* neurons (arrowheads)

within the hypothalamus. **g–g₁** Immunolabelling for pro-TRH in adult hypothalamus. **h** The morphology of ONECUT3⁺ neurons in GAD65-GFP and *Trh*-tdTomato reporter mice (arrowheads denote processes). **i** Scheme outlining differential neurotransmitter expression in hypothalamic territories, and the positions of ONECUT3⁺ neurons within. Blue-to-green gradient indicates a transition from GABA to glutamate territories. In situ hybridizations and immunohistochemistry were repeated at least two times. 3V third ventricle, AHA anterior hypothalamic area, ARC arcuate nucleus, DMH dorsomedial hypothalamus, E embryonic day, IHC immunohistochemistry, LH lateral hypothalamus, MZ mediolateral zone, P postnatal day, PeVN periventricular nucleus, SCN suprachiasmatic nucleus, TU tuberal nucleus, VMH ventromedial hypothalamus; scale bars = 100 μm (**b, b₁, d, e, g**), 20 μm (**a, a₁**, and insets in **b, b₁, c–h**).

cells/condition; Fig. 5c, d)[38,39], as well as their motility (Movies S1 vs S3). Transfected cells formed elaborate neurites with their lengths vastly exceeding those of mock-transfected cells (Fig. 5c, d₂). Acetylated-tubulin, labeling stable/long-lived neurites[40], as well as microtubule-associated protein 2 (MAP2), a somatodendritic marker for mature neurons[41], were both increased in the soma and neurites of transfected Neuro-2a cells (Fig. 5a and Supplementary Fig. 7a). These data suggest that ONECUT3 allows for a switch from proliferation towards differentiation.

Next, we opted for a glioblastoma cell line (U-251) to test the extreme possibility that ONECUT3 could induce a cell identity switch (glia → neuron). Indeed, ONECUT3 overexpression in U-251 human glioblastoma cells limited their proliferation, and even

induced the expression of beta-III-tubulin (TUJ1 antibody; Supplementary Fig. 7b, b1), a cytoskeletal marker primarily associated with immature neurons[42]. Even more unexpectedly, MAP2 was occasionally seen in the soma and nascent processes of *Onecut3*-transfected U-251 cells (Supplementary Fig. 7c). The ectopic expression of pro-neuronal markers paralleled the decrease of glial fibrillary acidic protein (GFAP), which characterizes astrocytes and their progenitors (Supplementary Fig. 7b, b₁).

Lastly, we derived neurospheres from E14.5 mouse cortices and hypothalami and maintained those for up to four days to test the above identity changes in a native cell system. Both transient transfection and lentiviral transduction of *Onecut3* resulted in a significant decrease in the cluster diameter of the neurospheres (149.6 ± 10.4 μm (control) vs

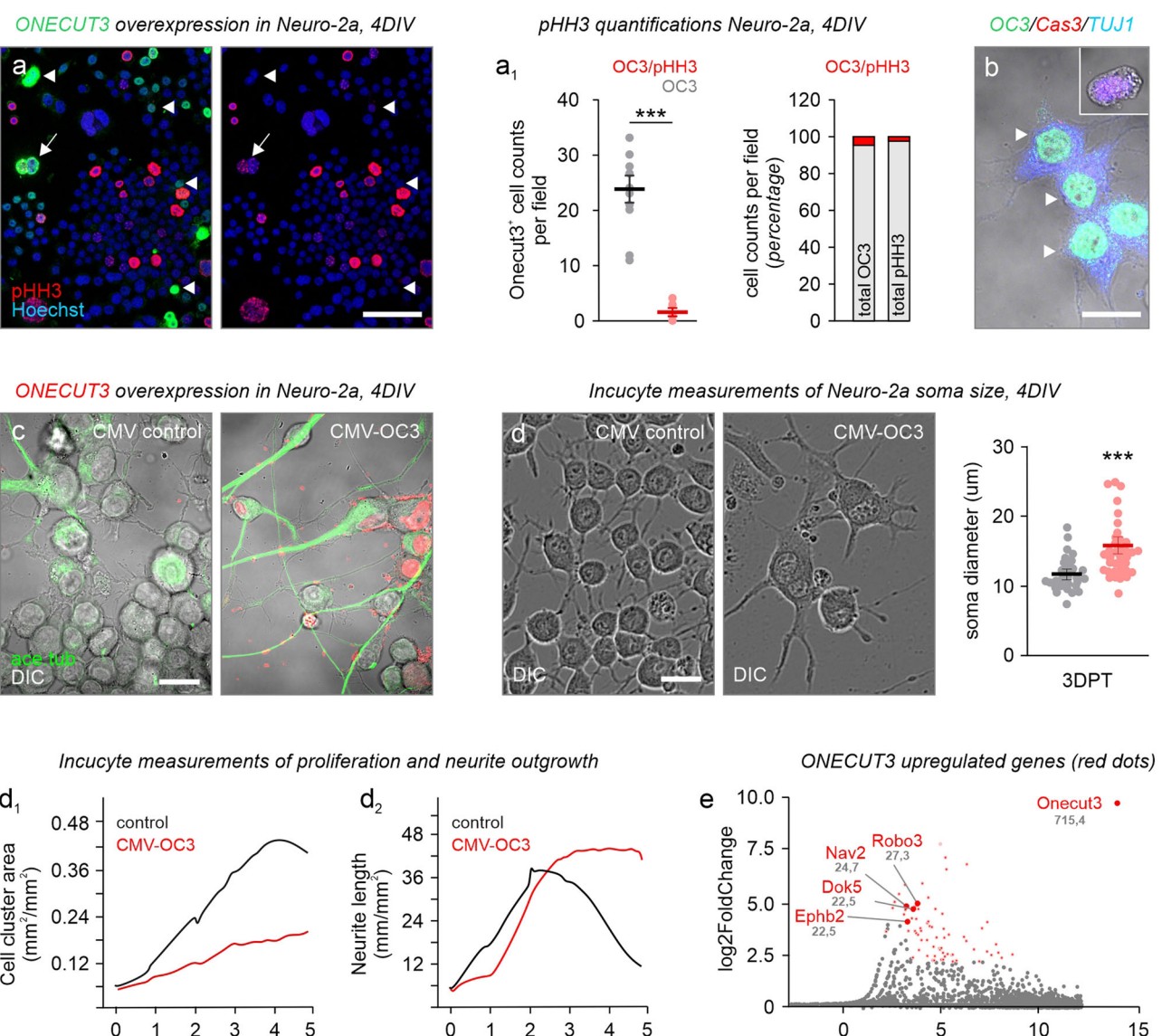

**Fig. 5 | ONECUT3 overexpression induces the differentiation of Neuro-2a cells.**
**a, a₁** Overexpression of ONECUT3 limited Neuro-2a cell proliferation as shown by a reduction in the mitotic marker pHH3 (*n* = 259 cells per condition over 11 fields imaged, *p* = 7,53E-07). **b** Cleaved caspase-3 (Cas3) was absent from transfected cells. **c** Neuro-2a cells three days post-transfection with mock (control; left) or CMV-*Onecut3* (OC3) plasmid (right). Note the induction of stable processes (acetylated tubulin⁺, green) upon ONECUT3 overexpression. **d** Representative images of Neuro-2a somata (differential interference contrast; DIC) after 4 days in vitro (DIV). Cell body size was measured by tracing the diameter of the somata, and analyzed with an unpaired two-tailed Student's *t*-test. (*n* = 47/group, *p* = 8,19508E-06). **d₁, d₂** Live-cell imaging of the cell cluster area (**b₁**) and neurite length (**b₂**) over a period of five days. **e** Dot plot illustrating relative gene expression as measured by sequencing bulk mRNA of Neuro-2a cells four days post-transfection. The top target genes involved in cytoskeletal remodeling were marked in red. Data were assessed with unpaired two-tailed Student's *t*-test and expressed as means ± s.e.m. ***$p < 0.001$ (**a₁, d**). Source data are provided as a Source Data file. ace. tub. acetylated tubulin, CMV cytomegalovirus promoter, DIV days in vitro, OC3 *Onecut3*. Scale bars = 100 μm (**a**), 20 μm (**b–d**).

89.5 ± 4.5 μm (transient transfection); *p* < 0.001 and 134.5 ± 8.0 μm (control) vs 98.5 ± 0.34 μm (transduction); *p* < 0.05; *n* > 100 neurospheres/condition). In all cases, ONECUT3⁺ cells were negative for pHH3 (Supplementary Fig. 8a, b). Furthermore, ONECUT3⁺ cells did not co-localize with SOX2, which marks neural stem cells in neurospheres[43], confirming their postmitotic state (Supplementary Fig. 8c). Thus, we suggest that ONECUT3 could promote a departure from proliferation towards neuronal maturation.

**Neuron navigator-2 is a downstream target of ONECUT3**
We used bulk RNA-seq of ONECUT3-overexpressing vs mock-transfected Neuro-2a cells (*n* = 3 biological replicates/group) to identify differentially expressed genes. A total of 911 genes out of 57,132

(including both protein-coding and non-coding genes, mouse genome release M35 (GRCm39−2020-06-24)) were classified as differentially regulated (Fig. 5e and Supplementary Fig. 9). Because ONECUT3 induced neuritogenesis in Neuro-2a cells, we then filtered genes whose expression increased > 4-fold (resulting in 62 genes at *q* < 0.01), and are implicated in neurite outgrowth and/or pathfinding as per gene ontology (GO) classification[20,21,44–46]. Out of the significantly altered genes (Fig. 5e, red), neuron navigator-2 (*Nav2*) was particularly relevant as a target (*q* < 0.001), because NAV2 interacts with TRIO to alter cytoskeletal dynamics, with its homolog, *sickie*, in *Drosophila* being directly controlled by the solitary *onecut* gene[47].

NAV proteins promote microtubule extension at plus-ends through their interaction with TRIO and subsequent activation of Rho GTPases

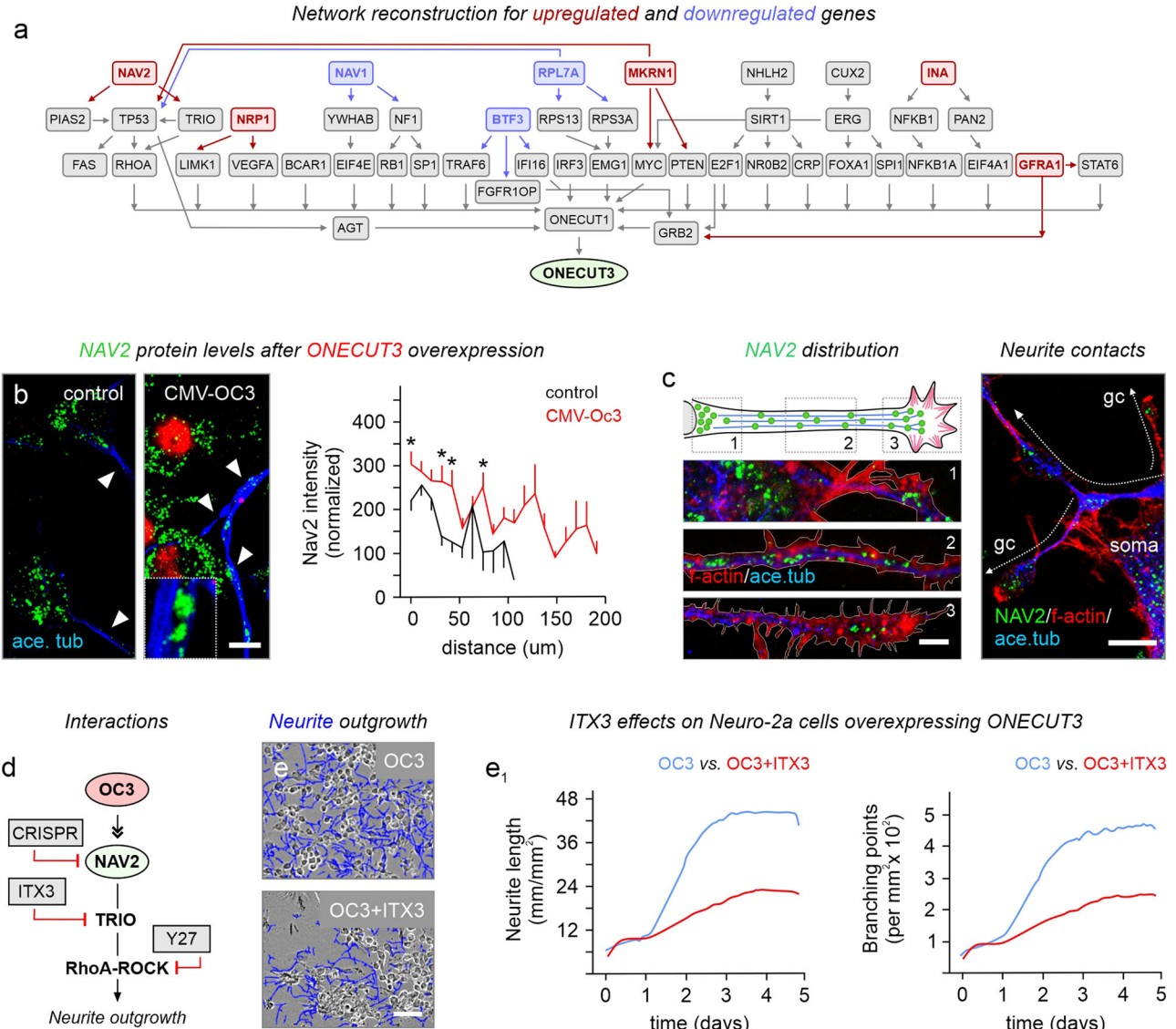

**Fig. 6 | NAV2 shapes ONECUT3-mediated neurite outgrowth in a TRIO-dependent fashion. a** Reconstruction of a protein-protein interaction network distinguished a pathway from ONECUT3 downstream to NAV2 within the signaling cassette supporting regulation of RhoA-mediated cytoskeletal dynamics. Note that arrows indicate an interaction, but not their directionality per se. **b** Changes in NAV2 protein levels after ONECUT3 overexpression. NAV2 signal intensity from the somata along neurites until their motile end tips were plotted. Data were acquired by Plot profiling in ImageJ ($n = 68$ cells/group). **c** NAV2 protein localization and distribution in the soma, along the process, and in pseudo-growth cones of Neuro-2a cells after ONECUT3 overexpression. Neuro-2a cells extended their neurites without forming classical synapse-like structures. **d** Pathway prediction downstream from ONECUT3, including NAV2 and its interacting partners known to

trigger cytoskeletal remodeling to promote neuritogenesis. **e, e₁** Neurite outgrowth and branching were inhibited by ITX3, a Trio N-terminal RhoGEF domain inhibitor. Experiments were performed in triplicate. Representative DIC images at 3 days post-transfection showed the reduced length of processes upon ITX3 treatment (**e**). Neurite length and branching were measured by IncuCyte live-cell imaging (**e₁**). A neurite mask (blue) was overlaid on the processes. Note that due to extensive proliferation, the difference in total neurite length was obscured in the first 2 days post-transfection until control cultures reached confluence (the experiment was performed twice). Data were evaluated by unpaired two-tailed Student's *t*-test. *$p < 0.05$. Source data are provided as a Source Data file. ace. tub. acetylated tubulin, CMV cytomegalovirus promoter, DIV days in vitro, gc growth cone, OC3 ONECUT3. Scale bars = 100 μm (**e**), 20 μm (**b, c**).

Rac1 and RhoG[48]. By using OmnipathR and KNIT[49,50] on open-label data for protein-protein interactions in mice, we reconstructed an interaction network linking ONECUT3 → NAV2 → TRIO → RhoA, the latter being a final effector of cytoskeletal remodeling (Fig. 6a)[51]. Indeed, we find NAV2 protein levels increased at the zonation of stabilized acetylated-tubulin and F-actin filaments along neurites, including pseudo-growth cone-like tips, in Neuro-2a cells transiently transfected with *Onecut3* ($p < 0.05$ measured at 10 μm, 40 μm, 50 μm, and 80 μm from the soma; Fig. 6b, c). Neurite outgrowth was not inhibited by physical contact with other cellular compartments, suggesting that increased NAV2 levels did not affect contact recognition (Fig. 6c). We then used a pharmacological

approach to confirm the intracellular signal transduction pathway (Fig. 6d and Supplementary Fig. 10a, a₁). Selective inhibition of TRIO by ITX3, a Trio N-Terminal RhoGEF Domain Inhibitor (50 μM), occluded Onecut3-induced neurite extension in Neuro-2a cells overexpressing *Onecut3* (Fig. 6e, e₁), as was shown by time-lapse live-cell imaging (note that cell viability was not affected; Fig. 6e and Supplementary Movies 1–4). These data suggest that ONECUT3 maintains a signaling axis that modulates cytoskeletal architecture for neuronal morphogenesis, at least in vitro.

Next, we recapitulated these findings in primary neurons derived from the hypothalami of E14.5 *Onecut3*-mCherry reporter mice ($n = \sim 4$

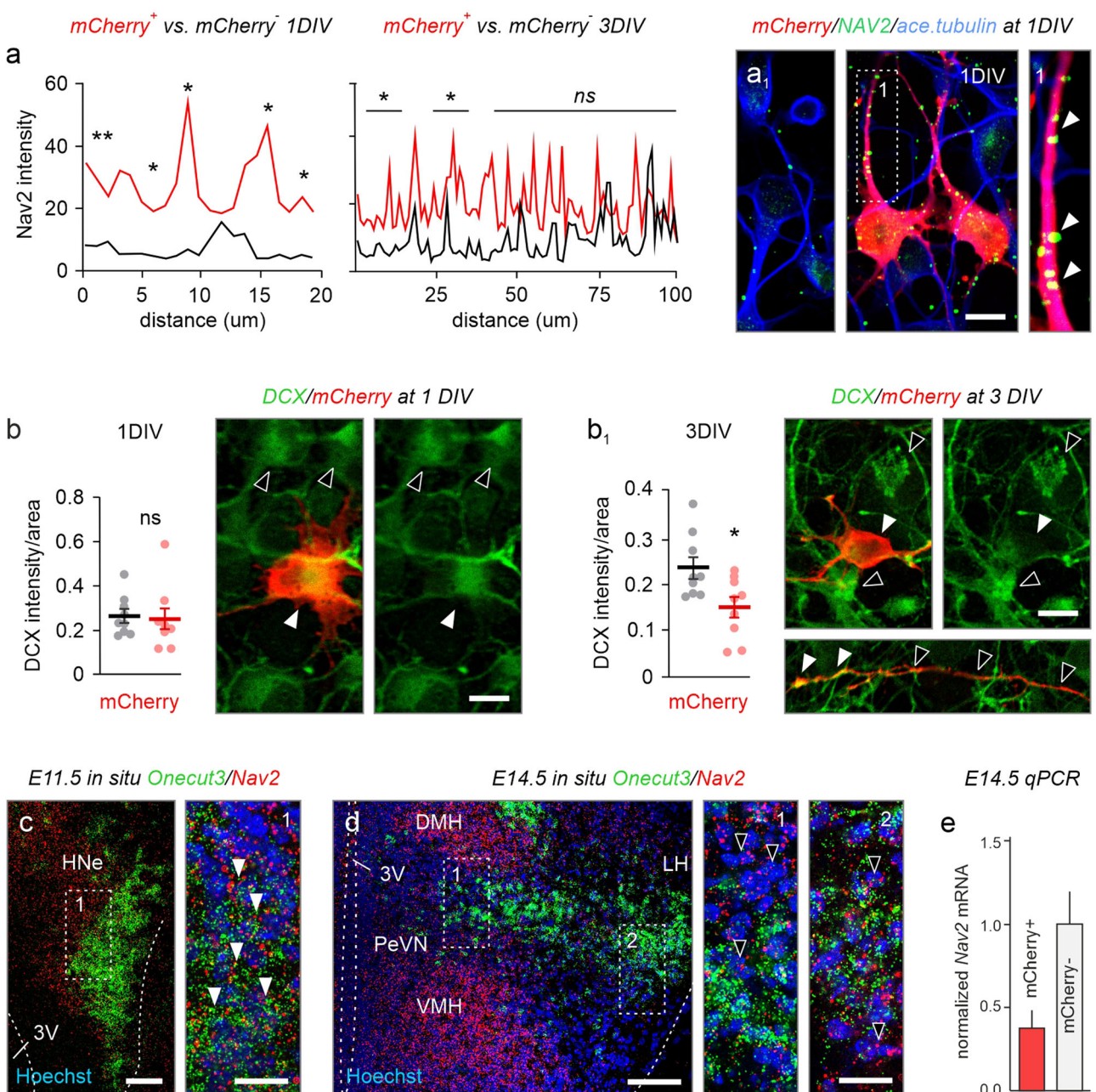

**Fig. 7 | Nav2 expression in developing hypothalamic neurons. a, a₁** NAV2 protein levels in primary hypothalamic neurons at 1 and 3 DIV. NAV2 levels were significantly higher in mCherry⁺ neurons from *Onecut3*-mCherry fetuses, as compared to other hypothalamic neurons at 1DIV (two-tailed Student's *t*-test). A lesser difference was seen at 3DIV. NAV2 signal intensity was quantified along neurites using Plot profiling in ImageJ (*n* = 11 cells/group for 1DIV, *n* = 12 cells/group for 3DIV; *p* < 0.05), as exemplified in a₁ (arrowheads mark NAV2⁺ substructures in the neurite). **b, b₁** DCX in mCherry⁺ hypothalamic neurons at 1DIV and 3DIV (open vs solid arrowheads). Differences in DCX expression were analyzed by quantifying immunofluorescence signal intensity within the soma (*n* = 9 for both 1DIV and 3DIV, *p* = 0.04). Data represent means ± s.e.m.; statistical differences between the groups were assessed by unpaired two-tailed Student's *t*-test. **c** In situ hybridization revealed the co-existence of *Onecut3* and *Nav2* at E11.5 in mice (arrowheads). **d, e** *Nav2* mRNA in *Onecut3*⁺ neurons in vivo at E14.5, as demonstrated by in situ hybridization (**d**) and qPCR (**e**) on mCherry⁺ (*Onecut3*⁺) hypothalamic neurons sorted from *Onecut3*-mCherry transgenic mice (*n* = 4 pooled embryos/sample) plotted as means ± s.e.m. In situ hybridizations were repeated at least two times. Source data are provided as a Source Data file. 3V third ventricle, DMH dorsomedial hypothalamus, E embryonic, HNe hypothalamic neuroepithelium, LH lateral hypothalamic area, PeVN periventricular nucleus, VMH ventromedial hypothalamus. Scale bars = 100 μm (**a, b**), 20 μm (**a, b₁**; insets in **c, d**), 10 μm (**b**).

subjects/experiment). Significantly higher amounts of NAV2 were partitioned along the neurites of mCherry⁺ neurons, as compared to their mCherry⁻ counterparts during the initial phase of neuronal polarization on day 1 in vitro (*p* < 0.05 at bins of 0–2, 4–6, 8–10, and 16–18 μm from the soma; *n* = 20 segments in *n* = 11–12 neurons/condition; Fig. 7a, a₁). After 3 days, NAV2 accumulated in distal neurites. Neither NAV2 immunoreactivity nor distribution in mCherry⁺ neurons was different from those in mCherry⁻ ones, suggesting that the induction of neuritogenesis is likely reliant on ONECUT3 → NAV2 signaling (Fig. 7a). At the same time, significant reduction in doublecortin (DCX), a microtubule-stabilizing protein expressed in immature and migrating neurons[52,53], was found in ONECUT3⁺ neurons (*p* < 0.001; *n* = 9 cells/condition; Fig. 7b, b₁).

In vivo, we found a similar reduction in *Nav2* mRNA in hypothalamic *Onecut3*⁺ neurons through mid-gestation (E11.5–E14.5) with

neuroblasts leaving the ventral proliferative zone containing elevated levels of *Nav2* mRNA at E11.5 (Fig. 7c). This was followed by a remarkable reduction in *Nav2* mRNA in ONECUT3[+] neurons undergoing neurite expansion by E14.5 (Fig. 7d). qPCR from mCherry[+] neurons FACS-ed from *Onecut3*-mCherry hypothalami on E14.5 confirmed this downregulation with lower *Nav2* mRNA levels (-37.5%, $n = 2$–3 embryos pooled) than in the non-labeled cell fraction (Fig. 7e). Yet, marked *Nav2* expression was shown in late-developing medial/ventricular territories (DMH/VMH), a cellular arrangement compatible with the outside-in plan of hypothalamic development[2]. Thus, we suggest that the ONECUT3 → NAV2 → RhoA cascade could prime neurons for cytoskeletal modifications and subsequent neuritogenesis.

## ONECUT3 loss-of-function disrupts neuronal morphogenesis in vivo

We first sought to test the in vivo significance of our findings by siRNA-mediated knockdown of *Onecut3* in the LH, which contains a dense population of glutamate/TRH[+]/ONECUT3[+] neurons[54]. An early postnatal time window was chosen for experimental manipulations, because (i) a custom-designed stereotaxic adapter allowed us to precisely inject siRNA[55], (ii) postnatal expression of ONECUT3 suggests a prolonged differentiation trajectory, and (iii) hypothalamic neurocircuits mature at infancy due to, e.g., hormonal priming[56]. Unilateral administration of *Onecut3*-targeting siRNA in the LH of *Onecut3*-mCherry pups at P4[55] significantly decreased ONECUT3 protein levels measured in the nuclei of mCherry[+] neurons 5 days post-injection ($n = 4$ animals/group; $p < 0.05$; Fig. 8a–a$_1$). The total number of *Onecut3*[+] neurons was not affected, indicating no effects on neuronal survival (Fig. 7a$_2$), which is in line with previous data showing that both control and targeting siRNAs do not induce neuronal cell death[57]. In situ hybridization revealed a significant decrease in both *Onecut3* and *Nav2* mRNAs (intensity *Onecut3*: $881 \pm 24$ vs $741 \pm 35$; intensity *Nav2*: $278 \pm 7.5$ vs $249 \pm 6.9$; $p < 0.05$; $n = 3$ animals/condition; Fig. 8b–b$_2$). However, we did not find a reduction in *Slc17a6* (VGLUT2) mRNA in ONECUT3-containing cells ($p = 0.356$). To reveal the effect of ONECUT3 loss on NAV2-mediated neurite outgrowth, we quantified mCherry[+] neurite coverage, encompassing both axons and dendrites, both harboring NAV2 (Fig. 6b, c). The density of mCherry[+] neurites in the LH was reduced relative to the contralateral hemisphere, which served as an internal control (fiber coverage $4.67 \pm 1.7$ % (control) vs $3.17 \pm 1.5$ % (siRNA); $p < 0.05$; $n = 4$ animals/group; Fig. 8c, c$_1$). These findings suggest that ONECUT3 could affect neuronal differentiation in the mouse brain.

Subsequently, we have chosen genetically modified *C. elegans* to reinforce the above findings. In *C. elegans*, developing neurons co-express *unc-53* (*Nav2* ortholog, which negatively regulates GTPases)[58] and *ceh-48* (*Onecut3* ortholog)[59], a neuronal transcription factor[60]. Given that *sickie*, the *Drosophila* orthologue to nematode *unc-53* and mammalian *Nav2*, is directly controlled by *onecut*[47], we hypothesized that *ceh-48* could equally regulate *unc-53* in *C. elegans*. By using qPCR on L4 larvae (to prevent interference due to egg production), we found significantly reduced *unc-53* mRNA in *ceh-48* loss-of-function mutants ($n = 3$ separate larval pools/genotype, $p < 0.05$; Fig. 9a). We then visualized neurites in wild-type (N2/WT), *ceh-48* (tm237), and *unc-53* mutants (mt152) by tetramethylindocarbocyanine perchlorate ('DiI'), which due to its lipophilic nature is absorbed at sensory cilia around the mouth into the dendrites of anterior amphid neurons[61]. Thus, the entirety of the amphid nerve could be visualized (Fig. 9b and Supplementary Fig. 11a). When analyzing *ceh-48* mutants, we first noted a significant increase in pharynx length, as compared to age-matched N2 wild-type worms (WT: $745 \pm 15.8$ μm vs *ceh-48*: $917 \pm 22.5$ μm vs *unc-53*: vs $775 \pm 24.5$ μm; $p < 0.001$; $n = 12$–14 worms/group; Fig. 9a, a–c and Supplementary Fig. 11c; measured from the tip of the nose until the end of the terminal bulb).

Particularly, amphid neurons in *ceh-48* mutants were misplaced, and their processes were deformed or even lacking (Fig. 9b$_1$, c$_2$). When present, the dendrites of amphid neurons were considerably longer than those in N2 wild-types (WT: $533 \pm 10.2$ μm vs *ceh-48*: $807 \pm 37.9$ μm vs *unc-53*: vs $663 \pm 30.6$ μm; $p < 0.001$; $n = 12$–14 worms/group Fig. 9c, c$_1$), yet defasciculated with many showing beaded ('varicose') appearance (Fig. 9b$_1$). We assumed that the increased length of amphid dendrites in *ceh-48* mutants reflected systems-level adaptation to the displacement of amphid neurons around the terminal bulb, with inner labial neurons surrounding the terminal bulb rather than segregating in the anterior direction (Fig. 9b, b$_1$, and c$_2$). Thus, *ceh-48* loss-of-function mutants exhibit organizational deficits in their inner labial sensilla, likely due to impaired neuronal morphogenesis[62].

In *unc-53* mutants, pharynx length was not affected significantly, even if this mutation impacted the general size of the worms (Supplementary Fig. 11b, c). Nevertheless, the loss of *unc-53* provoked deficits manifesting as (i) displaced terminal bulb (Supplementary Fig. 11b), (ii) the loss of amphid dendrites (Supplementary Fig. 11b), and (iii) their defasciculation and misplacement (neuronal location, $p < 0.001$; $n = 12$–14 worms/group; Fig. 9c$_1$, c$_2$ and Supplementary Fig. 11b). Thus, *unc-53* loss-of-function faithfully phenocopied that of *ceh-48*, suggesting that the two genes could jointly or coincidently drive organogenesis.

Since amphid neurons, the primary sensory neurons within the chemosensory circuit of *C. elegans*[63,64], exhibited morphological deficits in *ceh-48* mutants, we hypothesized that chemosensation might be impaired upon *ceh-48* loss-of-function. Therefore, we performed a chemotaxis assay with an attractant odorant (benzaldehyde) positioned unilaterally and calculated the chemotaxis index, with values closer to 1 indicating a strong attractant. The amount of *ceh-48*, as well as *unc-53* mutant worms that had arrived at the attractant area was significantly decreased, as compared to N2 wild-type controls (chemotaxis indices–N2: 0.56 vs *unc-53*: 0.38 vs *ceh-48*: 0.33; $p < 0.05$ (N2 vs *ceh-48*); $p < 0.001$ (N2 vs *unc-53*); $n = 3$ separate experiments; >400 worms per condition; Fig. 9d, d$_1$). Their general motility was not affected, as controlled by the ethanol test plate. In sum, these data from mouse and *C. elegans* models link *Onecut3*, *Nav2*, and their orthologs to the coordination of neuronal morphogenesis for function determination in neurocircuits.

## Discussion

Exploring the developmental organization of the hypothalamus lagged for decades behind the detailed analysis of dorsal forebrain structures[65–68] given its exceedingly intricate networks and complex synaptic and endocrine outputs from phenotypically and functionally segregated neuronal subtypes, as well as the lack of a general positional template. Experimental studies so far focused on matching TF signatures with positional cues[69] to resolve the primary transcriptional codes that distinguish discrete hypothalamic neuron populations, whilst placing them into the preoptic, anterior, tuberal, and mammillary areas[6]. A relatively recent expansion in single-cell RNA-seq studies additionally aimed at interrogating evolutionary variations in hypothalamic organization across animal species[3,18,59,70–72], which not only charted the many neuronal subtypes but also provided initial insights in wider gene regulatory networks (GRNs). Although the latest spatial transcriptomics of hypothalamic subregions provides precise TF expression vs positional information on even small subsets of neurons[69], there is a pressing gap of knowledge concerning the function of 'master genes' in GRNs, which converge onto instructing general neuro morphological changes during hypothalamus development. This is significant, as common hypothalamic progenitor pools (particularly those marked by the TF *Ascl1*)[7,9] can generate neurochemically distinct neuronal subtypes that differentiate along similar time scales.

*Postnatal mouse Onecut3 siRNA, IHC ONECUT3*

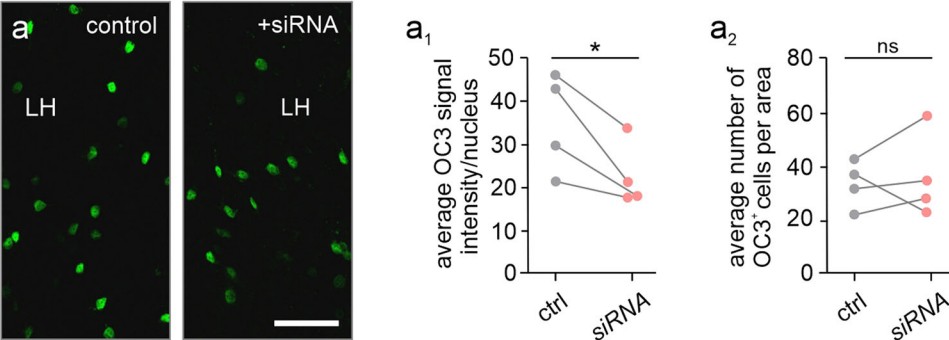

*Postnatal mouse Onecut3 siRNA, ISH Nav2/Onecut3*

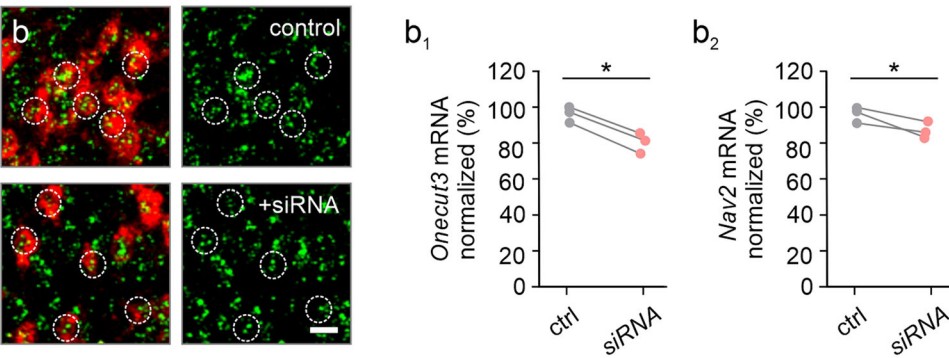

*Postnatal mouse Onecut3 siRNA, IHC mCherry⁺ fiber network*

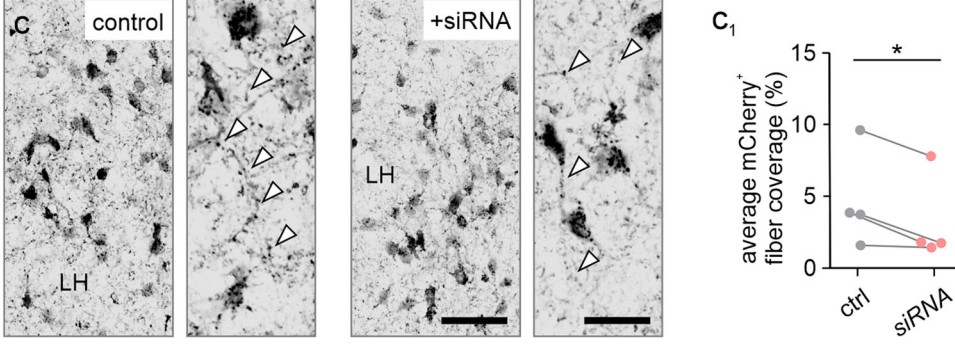

**Fig. 8 | ONECUT3 loss-of-function in the PeVN of the mouse hypothalamus.**
**a** siRNA-mediated *Onecut3* (OC3) knock-down reduced ONECUT3 protein levels in the nuclei (open arrowheads) of *Onecut3*-mCherry⁺ neurons in the LH of neonatal mice. Mice were unilaterally injected in the LH on P4 and sampled on P10. ONECUT3 protein content was visualized by immunohistochemistry. **a₁** ONECUT3 (OC3) signal intensity was compared between the non-injected (ctrl) vs siRNA-injected hemispheres of mice ($n = 4$/group; $p = 0.0431$). Data from each mouse were interconnected. **a₂** The number of ONECUT3⁺ neurons that populated the LH did not differ as a result of siRNA injection (ns non-significant; $n = 4$ subjects/group). **b–b₂** In situ hybridization revealed the downregulation of *Nav2* mRNA after siRNA treatment (dotted lines denote the individual cells measured, $n = 3$ subjects/group, 15 cells; $p = 0.0433$ (**b₁**) and 0.0363 (**b₂**). **c, c₁** Knock-down of *Onecut3* reduced neurite density, as measured by the area coverage of mCherry⁺ fibers in the LH, noting that *Onecut3*⁺ neurons were multipolar. The area occupancy of mCherry⁺ neurites, likely local dendrites, in the LH of *Onecut3*-mCherry mice, was compared between the non-injected (ctrl) vs siRNA-injected hemispheres ($n = 4$ mice; $p < 0.05$). Data from each mouse were interconnected. $p = 0.0497$, n.s. non-significant, two-tailed Student's *t*-test. Scale bars = 250 μm (**a**), 20 μm (**b**), and 100 μm (**c**). Source data are provided as a Source Data file.

Here, we provide evidence for an evolutionarily conserved GRN with equivalent consequences in both GABA and glutamate neurons derived from *Ascl1*⁺ progenitors. Unlike in other tissues where ONECUT TFs are purely accounted for promoting fate regulation[14,15,27,73,74], ONECUT3 is an indiscriminate TF labeling both postmitotic glutamate/TRH⁺ and GABA/TH⁺ neurons, which are spatially segregated into the LH and PeVN, respectively. These neuronal subpopulations are generated prior to E10.5 in the dorsolateral segment of the proliferative zone along the 3rd ventricle, and chain-migrate towards their final positions with the central contingent becoming GABAergic, while the lateral cell group acquiring a glutamate phenotype. As both populations, are embedded in regions mostly matching their own neurotransmitter identity, and glutamate, as well as GABA, can instigate neuronal differentiation and neurite outgrowth[75–77], we suggest that both neurotransmitters could act as homotypic chemotropic cues upstream of ONECUT3 to drive neuromorphological changes. For

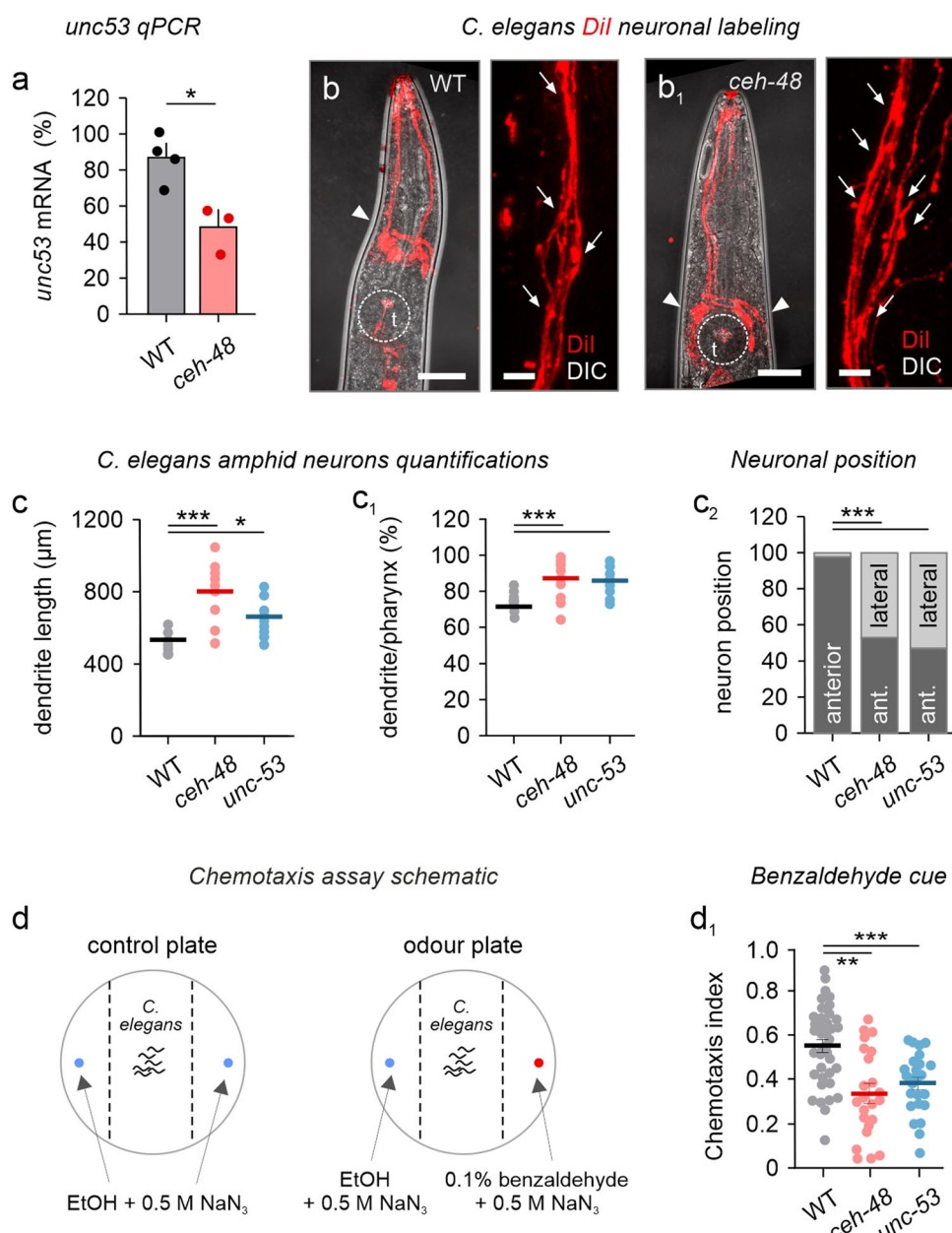

**Fig. 9 | Loss of *ceh-48 (Onecut* homolog*)* reduces dendrite complexity and chemotaxis in *C. elegans*. a** *Unc-53* mRNA levels from wild-type and *ceh-48* mutants (*n* = 3 (*ceh-48*) and 4 (WT) collections of larvae/group; *p* = 0.0191). **b, b₁** Wild-type (N2; WT) and *ceh-48* (tm237) *C. elegans* mutant worms were exposed to DiI to label their amphid neurons and their processes. Images were taken on a differential interference contrast background to demarcate anatomical structures, particularly sensory dendrites. *Ceh-48* knockout led to the repositioning of amphid neurons (arrowheads) in relation to the terminal bulb (t), and the defasciculation of their dendrites (arrows). **c–c₂** The length of amphid dendrites was increased in *ceh-48* and *unc-53* mutants, as compared to wild-type worms. For DiI uptake, *n* = 12 (WT) vs *n* = 16 (*ceh-48*) and *n* = 13 (*unc-53*) worms were used. *p* = 0.0000 and 0.0198 (**c**),

*p* = 0.0000 and 0.0007 (**c₁**), *p* = 0.0000 and 0.0062, 0.0058, 0.0092, and 0.0046 (**c₂**). Data were expressed as means ± s.e.m.; ***p* < 0.001, Student's *t*-test. **d** Schematic representation of the chemotaxis experiment with EtOH (control plate) and benzaldehyde odor (vs EtOH; experimental plate). NaN₃ was used to immobilize the worms. **d₁** Both c*eh-48* and *unc-53* mutants displayed reduced preference for benzaldehyde (experiments were performed in triplicates and pooled, *n* = 7184 (*ceh-48*), 9221 (*N2*) and 2459 (*unc-53*) worms analyzed; *p* = 0.001 (*ceh-48* vs *N2*), 0.0000 (*unc-53* vs *N2*), and 0.0769 (*unc-53* vs *ceh-48*). Data were expressed as means ± s.e.m. Statistical differences between the groups were tested by two-tailed ANOVA; **p* < 0.05, ****p* < 0.001 (post-hoc test). Scale bars = 75 μm (**b, b₁**), 20 μm (insert). Source data are provided as a Source Data file.

instance, glutamate exposure to cultured primary neurons can induce the TFs *cFos*, *Jun*, and *Zif268*, all involved in neuronal differentiation and brain development[78–80]. Thus, further research could decipher if local signaling cues are utilized to time ONECUT3 availability for the induction of neuronal differentiation, particularly cell-autonomous neurite outgrowth.

Members of the ONECUT family have been associated with cell fate determination and organogenesis[15,16]. Nevertheless, their downstream effectors specifying ONECUT-dependent physiological outcomes in the central nervous system remained elusive[13,81]. By using RNA-seq of ONECUT3-overexpressing Neuro-2a cells, we identified a network of downstream effectors, which are all implicated in axonal growth and guidance, including (i) roundabout receptor 3 (ROBO3), a substrate of netrin-dependent attraction[82], (ii) docking protein 5 (DOK5), an interacting protein associated with receptor tyrosine kinase-induced differentiation[45], (iii) ephrin B2 receptor (EphB2), mediating

ephrin-induced neurite retraction[46], and (iv) NAV2, which is associated with RhoA GTPase-dependent microtubule instability during neurite elongation and directional turning[20–22]. While we did not detect *Robo3* in *Onecut3*+ neurons in vivo, and *Ephb2* was ubiquitously expressed in the hypothalamus (see single-cell RNA-seq data in ref. 3), we found *Nav2* and *Onecut3* being co-expressed from E11.5 with their levels tailing-off when neurons will have reached their final positions by E14.5. Our cell culture data also support the notion that *Nav2* peaks upon the onset of differentiation (particularly neuritogenesis), consistent with its role in cytoskeletal remodeling[22] during cell maturation[83]. The importance of *Nav2* availability in promoting cytoskeletal remodeling during neuronal development is highlighted by the defasciculation and positional/survival phenotypes in *C. elegans*, by a reduction in hypothalamus size in *Nav2* knock-out mice[84], as well as the presence of brain malformations in human subjects carrying a biallelic truncated *Nav2* isoform[84].

While the loss of *Nav2* in mice results in strong discernable phenotypes[84], the effects of genetic ablation of ONECUT3 alone remain ambiguous. This is attributed to the similar homologs binding motifs of its family members ONECUT1 and ONECUT2, leading to compensation mechanisms, especially during critical developmental stages. Thus, existing ONECUT1/2 protein can similarly remodel chromatin[13] in the absence of ONECUT3, allowing downstream transcriptional events to resume as normal. As such, single mutants of the ONECUT family rarely demonstrate phenotypic changes as seen for instance in murine pancreatic cells[16], as well as in the *C. elegans* nervous system[85]. This genetic redundancy of not only paralogs, but also structurally unrelated proteins sharing similar functions, is a powerful evolutionary tool that prevents lethal phenotypes in the case of loss-of-function for critical genes[86,87]. For instance, genetic loss of the calcium-binding protein calbindin-D9k in mice does not induce any distinguishable phenotype, as if by itself would be insufficient to alter $Ca^{2+}$ levels. Instead, compensation by other $Ca^{2+}$ transporting proteins prevents a lethal phenotype to arise[88]. Likewise, a large discrepancy in zebrafish physiology is reported, where strong phenotypes induced by morpholino-based knockdown techniques cannot be replicated by full genetic knockouts due to compensation and upregulation of other genes[89]. Therefore, to circumvent this issue, we opted to use siRNA-mediated knockdown of ONECUT3 to observe if a strong reduction of ONECUT3 instead is sufficient to induce morphological. Indeed, we find a decrease of *Nav2* mRNA, as well as reduced neurite complexity of ONECUT3+ neurons, in regions exposed to siRNA. This finding is in line with the notion that disturbed mammalian ONECUT3 is worse than the complete loss of ONECUT3, as no single nucleotide polymorphisms in ONECUT3 in the human population are ascribed, indicating that any mutation that affects the functionality of this gene severely impacts survivability[3].

Surprisingly, the functional consequences of genetic *ceh-48* (ONECUT member) deletion in *C. elegans* were demonstrated as a direct phenocopy of the *unc-53* mutant through a strong reduction in *C. elegans* sensory transduction, possibly due to the misplacement of amphid neurons and their aberrant network integration. Comparably, genetic disruption of *both Onecut1/2*, leading to the complete loss of *Onecut3*, resulted in the dysregulation of diversification and distribution of spinal dorsal interneurons in the mouse[73], both necessary for the formation of sensory-motor connectivity patterns[90,91]. Taken together, our behavioral (chemotaxis) findings in *C. elegans* suggest the evolutionarily conserved control of sensory network development, with *Onecut3* as one possible central component of the relevant GRNs. In addition, the appearance of phenotypical changes in the *ceh-48* mutant suggests a lack of developmentally critical compensation mechanisms from either other distinct TFs or related ONECUT factors that diversified later in the evolutionary timescale in mammals.

Thus, a differentiation trajectory through *Onecut3* to precisely control the timely integration of select neurons into brain networks is provided here. We note that evolutionary conserved TFs should not only be seen as possible regulators of neuronal identity acquisitions alone, a focus of current large-scale single-cell mRNA screenings[3,6,71], but also as strong shared regulators of neuronal differentiation[13], analogous to those described in the developing peripheral system[17].

## Methods

### Ethical considerations on the use of live animals, and post-mortem tissues

Mice were kept under standard housing conditions in a humidity and temperature-controlled room with a 12-h/12-h dark/light cycle, and *ad libitum* access to food and water. Breeding and tissue collection conformed to the 2010/63/EU directive and was approved by the Austrian Ministry of Science and Research (66.009/0145-WF/II/3b/2014 and 66.009/0277-WF/V3b/2017). All experimental procedures were planned to reduce the suffering and numbers of the animals. For both anatomical mapping and in vitro experiments, adult mice (8–12 weeks), newborn pups, and embryos from timed pregnancies were bred on a C57Bl6/J background. For timed pregnancies, 1–2 female mice and a male were paired per cage, with the day of the vaginal plug designated as embryonic day (E) 0.5. The day of birth corresponded to postnatal day (P) 0. Transgenic mice used in this study were listed in Supplementary Table 1. The genotypes of mice were verified by DNA extraction from the tail (for embryos) or toe clips (for neonates) by incubation with 50 mM NaOH (Sigma; 600 μl/sample) at 95 °C for 10 min. Subsequently, samples were treated with 1 M Tris-HCl (Sigma, pH7.0; 50 μl/sample) and further processed by using the AccuStart II PCR supermix (VWR) and appropriate primer pairs (Supplementary Table 2). Reactions were performed in BioRad T100 thermocyclers.

Naked mole rats, Seba's fruit bats, and Indian flying foxes were obtained from Schönbrunn Zoo (Vienna, Austria) and their use was approved by the Austrian Ministry of Science and Research. Fetal wild boar and sheep tissue collection was approved by the German Centre for the Protection of Laboratory Animals, and processed by Simone Fietz and Wolfgang Härtig (University of Leipzig). Human tissues were obtained from the Brain Bank of the Institute of Neurology, Medical University of Vienna, Austria (head: Gábor G. Kovács). The use of fetal human brain samples was approved by the Ethical Committee of the Medical University of Vienna (Ethical approval number: 1316/2012). *C. elegans* was grown on standard nematode growth medium with *E. coli* (OP50) at 20 °C and maintained according to standard protocols[92]. Prior to the experiments, the worms were age-synchronized by egg-laying. The wild-type and transgenic strains used in this study are listed in Supplementary Table 3.

### Fixation and tissue processing

Whole embryos (up to E12.5), whole heads of embryos (E14.5), and freshly dissected brains (P3) were immersion fixed in 4% paraformaldehyde (PFA) in 0.1 M phosphate buffer (PB, pH 7.4) at 4 °C for 2–24 h (per size, *see below*) while being gently agitated by rotation. For older postnatal stages and adult brains, mice were transcardially perfused with the same ice-cold fixative, and post-fixed in 4% PFA in 0.1 M PB at 4 °C for 24 h. Subsequently, tissues were extensively rinsed in 0.1 M PB, and cryoprotected by immersion in 30% sucrose (in physiological saline) for at least 48 h before cryosectioning.

### Single-cell and bulk RNA-seq data analysis

Open-label scRNA-seq data (GEO accession number: GSE132730) were processed using the R program environment as previously described[93]. In brief, we have generated subset matrices for *Onecut3*+ expression, which contained parvocellular glutamate (including *Trh*+), GABA, and dopamine neurons on E15, E17, P0, P2, P10, and P23[3]. The Seurat R package (v4.0.6)[94] was used to analyze gene expression with variance

stabilizing transformation (sctransform v0.3.2)[95]. The identification of cell markers was based on matrices of log-normalized expression values with pseudocount and used to perform intersection-set analyses with the UpSetR R package (v1.4)[96]. Cell groups were visualized with a kernel-density estimation method in the Nebulosa R package (v1.4.0)[97].

Differential gene expression in bulk RNA-seq data after the in vitro overexpression of *Onecut3* in Neuro-2a cells and sequenced on Illumina HiSeq3000/4000 platforms was used to determine if and how *Onecut3* affected neuronal differentiation. A standardized Bioconductor pipeline[98] based on DESeq2 R package (v1.34.0)[99] was used with thresholds set as $p < 5e^{-3}$, base mean > 1, log$_2$FoldChange ≥ 2 (Supplementary Data 1). To address the likely mechanism of *Onecut3* action, the shortest path on a protein-protein interaction network in OmnipathR[49] was used (with select marker genes examined in both human and mouse databases), with KNIT[50] allowing the estimation of network topology for known effects of knock-out and/or knock-in experiments.

## qPCR

Total RNA was extracted from the mouse hypothalamus with an Aurum Total RNA kit (BioRad). cDNA libraries were generated by reverse transcription of the RNA samples using the High-Capacity RNA-to-cDNA Kit (Applied Biosystems). A total of 5–20 ng of cDNA was used for qPCR reactions (CFX Connect, BioRad) with SYBR Green master mix (Life Technologies). Primer pairs for mice and *C. elegans* were designed with Primer Blast (NCBI; Supplementary Table 4). Expression levels were normalized to tata-box binding protein (*Tbp*), a housekeeping gene.

*C. elegans* were picked from their plate (10 worms/sample), cleaned in water to remove residual bacteria, and incubated in 50 μl lysis buffer solution (5 mM Tris-HCl (pH 8.0), 0.5% Triton X-100, 0.5% Tween 20, 0.25 mM EDTA, 1 mg/ml proteinase K (all from Sigma), heated to 65 °C for 10 min. Proteinase K was inactivated by incubation at 85 °C for 1 min, followed by cooling the samples on ice. cDNA was synthesized with the Maxima H Minus cDNA synthesis kit (Thermo Fisher). Quantitative real-time PCR (CFX Connect, BioRad) was performed by using SYBR Green master mix (Life Technologies) and custom-made primer pairs (Primer Blast, NCBI). Expression values were normalized to peroxisomal membrane protein *pmp-3*, a housekeeping gene (Supplementary Table 5).

## In utero electroporation

Timed pregnant mice (E13.5) were anesthetized with an intraperitoneal injection of ketamine (90 mg/kg, MSD Animal Health) and xylazine (4.5 mg/kg, Ani Medica), and their abdominal cavity opened to expose the uterus. One of the uterine horns was carefully extracted by a ring forceps, and placed on a wet sterile surgical pad. A flamed capillary pipette containing 1–1.5 μl of colored DNA solution was slowly injected into the lateral ventricle of an embryo. The DNA solution was prepared as a mixture of pCAGGS-mCherry plasmid (final concentration of 1–2 μg/μl, donated by Katsuhiko Tabuchi) and 0.1% FAST green (Sigma) in phosphate-buffered saline (PBS, Sigma). The injection procedure was repeated for all embryos except for the last one at the end of the uterine horn. After waiting 2–3 min for sufficient DNA diffusion from the lateral ventricle to the 3rd ventricle, an electric impulse of 30 V was applied through a Nepagene electroporator (NEPA21) to electroporate the plasmid in the progenitor cell layer that lines the wall of the 3rd ventricle. Subsequently, the uterine horns were soaked in sterile PBS (Sigma), and carefully placed back into the abdominal cavity. Muscle and skin incisions were sutured, and the dams were placed on a heating pad (35 °C) to aid their recovery. Electroporated embryos were let to develop until they reached the designated ages, collected, and processed for immunohistochemistry.

## In situ hybridization

Whole embryos (<E14.5), whole heads (≥E14.5), and extracted brains (>P0) were flash-frozen and sectioned with a CryoStar NX70 Cryostat (Thermo Fisher) at 16 μm thickness. Sections were collected on SuperFrost⁺ glass slides (Thermo Fisher Scientific), and stored at −80 °C until processing. Tissue sections were pre-treated with 4% PFA at 4 °C for 20 min, washed in PBS, and dehydrated in an ascending gradient of ethanol (25%, 50%, 75%, and 100%; 5 min each). In situ hybridization was performed according to the HCR v3.0 protocol for *'generic sample on the slide'* with probe sets of *Onecut1, Onecut2, Onecut3, Trh, Th, Nav2, Slc17a6,* and *Slc32a1* (Molecular Instruments). Sections were imaged on an LSM880 confocal microscope (Zeiss), and processed with the ZEN software (Zeiss).

## Fluorescence immunohistochemistry

Immunohistochemistry was performed on 20 μm-thick cryosections (for ages up to P3) or 50 μm-thick free-floating sections. Specimens were washed with PBS (Sigma), and incubated with a blocking solution containing 5% normal donkey serum (NDS, Jackson ImmunoResearch), 2% bovine serum albumin (BSA, Sigma), and 0.2% Triton X-100 (Sigma) in PBS at 22–24 °C for 1 h. Next, tissues were exposed to combinations of primary antibodies (Supplementary Table 6) diluted in 2% NDS, 0.1% BSA, and 0.2% Triton X-100 in PBS at 4 °C for 72 h. After extensive washing, appropriate combinations of secondary IgGs conjugated with carbocyanine (Cy)2, 3, or 5 (raised in donkey, 1:300, Jackson ImmunoResearch) were applied at 22–24 °C for 2 h. Hoechst 33,342 (1:10,000, Sigma) was routinely used as a nuclear counterstain. After repeated rinses in PBS, sections were dipped in distilled water, air-dried, and coverslipped with Entellan (in toluene, Merck).

## Whole-mount immunofluorescence

Embryos were collected from time-mated pregnant mice and immersion fixed in 4% PFA (1.5 h for E8.5 and E9.5, 2.5 h for E10.5) at 4 °C. After washing in PBS−0.1% Tween-20 (Sigma, 3×) at 22–24 °C, embryos were further immersed in increasing concentrations of methanol (25%, 50%, 75%, and 100% methanol, 1 h each) at 22–24 °C. To reduce background staining, embryos were bleached in a solution containing 1/3 H$_2$O$_2$ and 2/3 20% dimethyl sulfoxide (Sigma)/80% methanol (termed 'Dent's fixative') at 4 °C for 24 h. After washing in 100% methanol, embryos were immersed in Dent's fixative (4 °C) for another 24 h. Following washes with PBS/Tween-20, samples were incubated in a cocktail of primary antibodies diluted in blocking solution (5% NDS, 20% DSMO, 75% PBS-Tween) at 22–24 °C for 7 days. Next, samples were incubated with appropriate secondary antibodies in a blocking solution at 22–24 °C for another 3 days. Samples were subsequently washed with PBS-Tween (6×, 30 min each), followed by 50% methanol/PBS (5 min) and 100% methanol (3×, 20 min each). Before imaging on a LSM880 confocal laser scanning microscope (Zeiss), embryos were cleared in BABB (benzyl alcohol/benzyl benzoate; 1:2). Three-dimensional reconstruction of whole embryos were performed in the ZEN software (Zeiss) by using a z-stack and tile-scan mode, and further processed with Imaris X64 9.0.2 (Bitplane).

## Brain-wide tissue clearing and light-sheet microscopy

Brains from Onecut3-mCherry embryos (E14.5) were used. Samples were fixed in 4% PFA at 4 °C overnight, incubated in 3% H$_2$O$_2$ (in PBS) for 24 h, and transferred into 30% sucrose in PBS (Sigma) at 4 °C for at least 48 h. They were then immersed in 'CUBIC 1 solution' (25% urea, 25% N,N,N′,N′-tetrakis-(2-hydroxypropyl)ethylenediamine, and 15% Triton X-100; all from Sigma) at 37 °C for 48–72 h. After repeated washes in PBS at 22–24 °C, the specimens were incubated in a solution of 2% BSA, 5% NDS, 0.5% Triton X-100, and 10% DMSO (all from Sigma) to block non-specific immunoreactivity during 3–6 h. Next, samples were immersed in a cocktail of primary antibodies (Supplementary Table 6) diluted in 2% NDS, 0.1% BSA, 0.3% Triton X-100, 5% DMSO, and

0.1% NaN$_3$ (Sigma) in PBS at 37 °C for 4–6 days. After extensive washing in M PBS, tissues were exposed to a mixture of Cy2-, Cy3- or Cy5-conjugated secondary antibodies (1:400, raised in donkey, Jackson ImmunoResearch) that had been diluted in 3% NDS, 0.1% NaN$_3$ in PBS. After 3 days at 37 °C, washing in PBS (4×, 30–60 min each) ensued. Samples were then immersed in 'CUBIC 2 solution' (50% sucrose, 25% urea, 10% 2,2',2''-nitrilotriethanol, 0.1% Triton X-100; all from Sigma) at 22–24 °C for 24–36 h. Samples were imaged with a Zeiss R.1 Lightsheet microscope while immersed in CUBIC 2 solution (refractory index = 1.45). Whole-brain images were acquired with z-stack and tile scanning modules in the ZEN software (Zeiss), and post-processed with Imaris X64 9.0.2.

## Overexpression of *Onecut3* in Neuro-2a and U251 cells
Neuro-2a (ATCC) and U251 (ATCC) cells were propagated in DMEM (4.5 g/l glucose, supplemented with GlutaMAX, Gibco) and containing 10% fetal bovine serum (FBS), 1 mM sodium pyruvate, 1 mM non-essential amino acids, 100 U/ml penicillin, and 100 μg/ml streptomycin (all from Gibco). Cells were transfected with 500 ng pDNA (CMV-*Onecut3* or CMV-SERT as CMV control) using a jetPRIME transfection reagent or a Nucleofector (Lonza Biosciences). The medium was then replaced with fresh growth medium (containing 2% FBS), and cultured for another 3 days. For immunocytochemistry, cells were plated on poly-D-lysine (Sigma)-coated glass coverslips at a density of 20,000 cells/well in a 24-well format 24 h prior to transfection. For live cell imaging (Incucyte SX5, Sartorius), Neuro-2a cells were plated at a density of 20,000 cells/well without glass coverslips, and imaged for up to 4 days post-transfection. For RNA sequencing, Neuro-2a cells were plated in a 6-well format at a density of 500,000 cells/well, and transfected with 1.5 μg DNA/well. Cells were lysed 3 days post-transfection, with their RNA extracted for bulk RNA-seq (see ref. 3 for details) and qPCR validation (Supplementary Table 7).

## Cultured neural stem cells and their *Onecut3* transduction
Cortices and hypothalami of E14.5 embryos were dissected out and dissociated in 0.1% trypsin-containing DNA in DMEM. After inhibition of trypsin, cells were washed extensively with DMEM and plated into NUNC-coated T75 flasks to allow the formation of neurospheres in full KnockOut medium (Gibco) for up to three days. The sequence-validated protein coding region (CDS) of the mammalian *Onecut3* was cloned into a pRRL-EF1a-rtTA3-P2A-ZeoR-IRES-GFP vector (provided by J. Bigenzahn, Center of Molecular Medicine, Austrian Academy of Sciences, Vienna, Austria)[100] using the *BsiWI* and *XhoI* restriction sites, thus replacing the rrTA3-P2A-ZeoR cassette. For transduction experiments, neurospheres were dissociated with 0.1% trypsin containing DNAse, cells plated in full KnockOut medium in NUNC-coated tissue culture dishes together with the lentivirus particles and LentiBOOST (Sirion Biotech). After 2-3 days of incubation, neurospheres were collected, fixed in 4% PFA in 0.1 M PB, and processed for immunohistochemistry.

## Primary hypothalamic neurons
Primary cultures of hypothalamic neurons were obtained from both wild-type and *Onecut3*-mCherry mouse hypothalami at E14.5 by using a Papain dissociation system kit (Worthington, PDS Kit LK003150). Briefly, embryos were extracted from the uterus and collected in ice-cold Hank's balanced salt solution (HBSS). Tissues were further dissociated mechanically using flamed glass Pasteur pipettes with decreasing tip sizes until a uniform cell suspension was achieved. Dissociated cells were centrifuged at 300 g for 5 min. The cell pellet was resuspended in an albumin-ovomucoid inhibitor/DNase solution to inhibit papain activity (Worthington). Cell suspensions were layered on top of the albumin-ovomucoid inhibitor and centrifuged at 70 g for 6 min to remove debris. Pellets were subsequently resuspended in Neurobasal A medium (Fisher Scientific), supplemented with Glutamax

(Thermo Fisher), 1% FBS (Life Technologies,), 2% B27 supplement (Gibco), and 1 μM Q-VD-OPh (Sigma), a caspase-3 inhibitor used to limit apoptosis[101]. To determine *Nav2* mRNA levels, mCherry$^+$ neurons were collected with fluorescence-activated cell sorting and processed for mRNA extraction (see above). For immunocytochemistry, neurons were plated on ploy-D-lysine-coated 96-well plates (Nunc MicroWell 96 optical bottom plates, Sigma) at a density of 80.000 cells/well. Neurons were either followed with an Incucyte SX5 imaging system (Sartorius) for up to 4 days, or processed for immunocytochemistry and imaged with an LSM880 confocal laser scanning microscope (Zeiss). For bulk mRNA extraction, neurons were plated in 24-well format at a density of 450,000 cells/well and lysed after 1 or 4 DIV.

## Live-cell imaging
Neuro-2a cells and primary neurons were imaged using a 20× objective for up to 4 days in an Incucyte SX5 system (Sartorius). Cell cluster area, neurite length, and neurite branching were analyzed over the observation period, and plotted by the proprietary software of Sartorius. We note that due to proliferation and the innate minor neurite lengths of Neuro-2a cells, the difference in total neurite length is obscured in the first two days post-transfection until appropriate confluence is reached in control cultures.

## Immunocytochemistry
Coverslips with adherent cells were washed with PBS (Sigma), fixed in 4% PFA in PBS for 30 min on ice, and incubated in a blocking solution (10% NDS, 5% BSA, 0.2% Triton X-100 in PBS) at 22–24 °C for 1 h. Next, cells were incubated with cocktails of primary antibodies (diluted in 5% NDS, 2% BSA, 0.2% Triton X-100 in PBS) at 4 °C for 24 h, extensively washed with PBS, and exposed to secondary antibodies conjugated with Cy2, Cy3 or Cy5 (1:300, made in donkey, Jackson ImmunoResearch) at 22–24 °C for 2 h. After several more washing steps with PBS, coverslips containing cells were removed from the wells, immersed in distilled water, and mounted with glycerol-gelatine (GG1; Sigma). Imaging of the cells was performed on an LSM 880 confocal microscope (Zeiss).

## DiI staining in *C. elegans*
Age-synchronized young adult worms were washed off their plates using an S-basal medium, followed by repeated washed to remove residual bacteria. Pelleted worms were incubated with DiI (5 μM solution, Cell Tracker CM-DiI Dye #C700) for 3 h in the dark, followed by repeated washes with S-basal medium[61]. Prior to imaging, worms were anesthetized using 100 mM NaN$_3$ and mounted on 1% agarose on glass slides. Orthogonal image stacks were acquired on an LSM880 confocal laser scanning microscope (Zeiss). Maximum intensity projections of representative images were shown. Quantification was performed using ImageJ 1.54 h.

## Chemotaxis assay
Prior to the chemotaxis assay, a drop of 0.5 M NaN$_3$ was added to the opposite end of the agar Petri dish to immobilize the worms once they reached the site. A drop of 0.1% benzaldehyde (in ethanol, Sigma) or ethanol was placed on the opposing sides of the plate, at the same spot as for NaN$_3$. Age-synchronized worms were washed several times in S-basal medium, and then placed in the middle of the plate (approximately 200 worms per condition). Agar plates were covered in parafilm, and incubated at 20–24 °C in the dark. After several hours, when the majority of the worms had traveled to either side of the plate, the number of worms that crossed a distance marker on the plate towards either odorant (*see* Fig. 9d,d$_1$) was counted. Each strain (wild-type vs both *unc-53* and *ceh-48* mutants) was assessed under control conditions (the plate only contained ethanol on both ends) and on experimental plates, that contained benzaldehyde vs ethanol on their opposing ends. A chemotaxis index was calculated as $C_i$ = ((number of

worms that crossed the odorant side)) – (number of worms on the ethanol side))/(number of the total amount of worms counted).

### siRNA-mediated knockdown

*Onecut3*-mCherry mouse pups of both sexes at P4 were anesthetized with isoflurane and placed in a stereotaxic frame (Kopf) with a custom-made gas dispenser[55]. The skull of each mouse was exposed by an incision through the skin. A 25 G needle was used to remove a small skull fragment to expose the surface of the brain. A stereotaxic injector with a glass capillary was used for the unilateral delivery of 250 nl siRNA (500 μM) targeted against *Onecut3* mRNA (Dharmacon, #SO-3101441G) at the following bregma coordinates: AP = −0.85 mm, *L* = −0.5 mm, and DV = −4.25 mm/−4.50 mm (two infusion of 125 nl each). The glass capillary was slowly withdrawn 5 min after the injection. The incision site was stitched, and the pups returned to the dams in their home cages. Mice were sacrificed 5 days later (at P10), and processed for immunohistochemistry as above. For quantification, 3–5 sections of the target area were analyzed per subject, and compared to the contralateral (control) hemisphere. Quantification of neurite area coverage per field and *Onecut3/Nav2* mRNA intensity per cell, was performed using ImageJ 1.54 h.

### Statistics

Data were expressed as means ± s.e.m. Data were analyzed using appropriate ANOVA designs. In histochemical experiments, Dunnett´s *post-hoc* test for multiple comparisons was used with a reference age or genotype. Otherwise, data were evaluated using Student's *t*-test (two-tailed, unpaired). Prism 8 (GraphPad) was used for analysis, with $p < 0.05$ considered statistically significant.

### Reporting summary

Further information on research design is available in the Nature Portfolio Reporting Summary linked to this article.

## Data availability

The snRNA-seq data generated in this study have been deposited in the NCBI Gene Expression Omnibus database under accession code GSE132730 and as 'Supplementary Data 1 - Neuro2a-onecut3-over-expression'. All other primary data were made publicly available at https://github.com/Harkany-Lab/Zupancic_2023 and deposited to Figshare.com with https://doi.org/10.6084/m9.figshare.22680433. Source data are provided with this paper.

## Code availability

Analysis of bulk RNA-seq data relied on published protocols[3,18].

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

## Acknowledgements

The authors thank Z. Máté, G. Szabó, and F. Erdélyi for the custom generation of transgenic mouse lines, C. Fekete for Trh transgenic tissues (all from the Institute of Experimental Medicine, Hungarian Academy of Sciences, Budapest, Hungary), A. Goudmaeker for IVF recovery of a frozen mouse line (SSS animal facility, Université catholique de Louvain), and Y. Yanagawa (Department of Genetic and Behavioral Neuroscience, Gunma University Graduate School of Medicine, Maebashi, Japan) for providing GAD67gfp/+ mice. We also thank S. Cloer, D. Preininger, and A. Weissenbacher (Tiergarten Schönbrunn, Vienna, Austria) for providing naked mole rats, Seba's fruit bats, and Indian flying foxes, as well as F. Aujard (CNRS, UMR 7179 'Adaptive mechanisms and evolution', France) for Microcebus tissues. I. Milenkovic and G.G. Kovács (Clinical Institute of Neurology, Medical University of Vienna, Vienna, Austria) are acknowledged for providing post-mortem human brain samples. We are indebted to S. Rehman (Medical University of Vienna), M. Kalusa (University of Leipzig, Leipzig, Germany), and W. Reimann (Paul Flechsig Institute for Brain Research, Leipzig, Germany) for their technical assistance. C. elegans strains were provided by the National Bioresource Project for the nematode, Japan, and the CGC, with the latter being funded by the NIH Office of Research Infrastructure Programs (P40 OD010440). This work was supported by the Austrian Science Fund (FWF, P 34121-B; to E.K.), the Swedish Research Council (2023-03058, T.Ha; 2020-01688, T.Hö.), the Swedish Brain Foundation (Hjärnfonden, FO2022-0300, to T.Ha.), the Novo Nordisk Foundation (NNF23OC0084476, to T.Ha.), the European Research Council (FOODFORLIFE, ERC-2020-AdG-101021016; to T.Ha.), the Université Catholique de Louvain ('Fonds spéciaux de recherche'-FSR, to F.C.), and Fonds de la Recherche Scientifique F.R.S.-FNRS ('Project de recherche (PDR)' #T.0039.21, to F.C.). S.J.E. is supported by the Simons Foundation #543069. I.L. is supported by a post-doctoral fellowship from the Human Frontiers Science Program (LT000335/2020-L). E.R. holds a PhD grant from the FRIA (F.R.S.-FNRS, Belgium). F.C. is a Research Director of the F.R.S.-FNRS (Belgium).

## Author contributions

E.K., T.Hö., and T.Ha. designed experiments, M.Zu., E.K., E.T., L.E., P.B., E.R., S.E., I.L., M.Z., E.R., S.F., and W.H. performed experiments. E.K., T.Ha., T.Hö., F.C., and A.V. procured funding. M.Zu., E.K., and T.Ha. wrote the manuscript. All authors commented on the manuscript and approved its submission.

## Funding

## Competing interests

The authors declare no competing interests.
