## [Peer Review File · Nature Communications]

Concerted transcriptional regulation of the morphogenesis of hypothalamic neurons by ONECUT3REVIEWER COMMENTS

Reviewer #1 (Remarks to the Author):

This intriguing paper from the Harkany's laboratory uncovers a role for the transcription factor *Onecut3* in facilitating the transition from immature to differentiated neurons via the control of an evolutionarily conserved transcriptional axis involving the neuron navigator-2 gene during embryogenesis and early postnatal development.

To reach their conclusions, the authors used a wealth of experimental approaches, including single-cell and bulk RNA-seq data analyses, comparative anatomy in mammals, including humans, genetic approaches in vivo both in mice and worms, as well as in vitro studies in neural cell lines.

My comments are as follows:

1. In the title of the article, why did the authors choose to mention "dopamine and glutamate neurons in the hypothalamus" and not dopamine and TRH neurons or GABA/tyrosine hydroxylase (TH) and glutamate/TRH to be more accurate?
2. In contrast to TRH neurons that appear to express exclusively vesicular glutamate transporter 2, couldn't TH neurons, which are tagged by the authors as being GABAergic, also be glutamatergic? Figure 2c appears to strongly suggest this.
3. Did the authors assess for potential sexual dimorphisms? For example, it is well-known that, in the anteroventral periventricular region of the preoptic region, the TH neuronal population is sexually dimorphic. Could this sexual dimorphism contribute to the loss of *Onecut3* expressing cells during postnatal development (e.g., Figure 1j)?
4. Throughout the manuscript, including page 4 and page 7, the terminology used by the authors to identify stages of postnatal development is erroneous. Indeed, P4 does not correspond to infantile mice, but neonatal mice. Similarly, P35 C57/Blk6 mice are far from being adult mice, as they actually are pre-pubertal. Postnatal development is usually described as follows: P0-P7 the neonatal period, P8-P20 the infantile period (when minipuberty occurs), P21-P35 the juvenile period, followed by the peripubertal period (puberty usually occurs around P45). Adulthood is only reached at P60.
5. Figure 7 is not very convincing (both a and b panels). Could the authors show the mCherry expression/immunolabelling? Could the authors provide fluorescent images with a wider field of view to see both targeted and non-targeted areas by the siRNA?
6. In the *Onecut3* knockdown experiments, was *Nav2* expression downregulated, as shown for *C. Elegans* Figure 8b?
7. Figure 8. Not being familiar with *C. Elegans* neuroanatomy, this reviewer has difficult times to understand what was measured, e.g., dendritic length. Could the authors provide a schematic illustrating what they actually measured? In the results section they speak of dendritic length while in the discussion/legend they mention defasciculation...? Where are the cell bodies of the neurons? In the WT panel, one can see bilateral symmetric projections/dendrites, while in the *ceh-48* panel projections/dendrites appear to be unilateral, is this real or proper to the selected image? If real, could it be due to an alteration of the expression of guidance molecules or of their receptors? Defasciculation could be also be a result of the alteration of the chemotrophic pathways (e.g. *Robo3* or *Ephb2* orthologs).

Reviewer #2 (Remarks to the Author):

This study addresses the expression and function of the *Onecut3* gene in the mouse hypothalamus

and the *C. elegans* nervous system. In the mouse, they find that *Onecut3* is expressed in neuroblasts derived from *Ascl1*-expressing progenitors, and maintained in subsets of neurons. Overexpression of *Onecut3* in cell culture and knock-down *in vivo* points to a role for *Onecut3* in regulating neuronal differentiation, specifically in promoting neurite extension, possibly by regulating a Nav2-TRIO-RhoA cascade. Complementary experiments in *C. elegans* on the orthologue *ceh-48* points to a related role for this gene in controlling neuronal differentiation in *C. elegans*. The gene expression analysis in the mouse hypothalamus is detailed and will be of value to the field. However, the functional analysis lacks rigour, is not entirely logical, and relies on siRNA and overexpression in cell lines. The absence of mouse knockout analysis is also a shortcoming. The following issues, if addressed, would help strengthen the manuscript.

Major issues

1. *Onecut3* loss-of-function analysis in the mouse is based upon siRNA, which is notorious for giving only partial knock-down and for off-target effects. *Onecut3* mouse mutant analysis would have been much better. If they do not have access to *Onecut3* mouse mutants, analysis of additional siRNAs and/or transient Crispr/CAS9 knockout experiments would greatly strengthen the manuscript. Furthermore, are there neuronal cell lines expressing *Onecut3* where its function could be probed by knock-down/knock-out?
2. The manuscript does not articulate what specific neuronal property/properties *Onecut3* controls? All neurons extend dendrites and axons, so this neuronal cellular polarity is not a feature specific to *Onecut3* neurons. Do *Onecut3* neurons project axons and/or dendrites to a common target(s)? Do they have unusually long/short dendrites/axons? Do they connect to a common type of pre/post-synaptic neuron? Similarly, what is the common evolutionarily conserved role of *Onecut3/ceh-48*? They state that both genes control "neuronal morphogenesis", but this is a rather broad phenotype.
3. Fig 7: Does *Onecut3* knock-down result in changes in neurotransmitter expression?

Minor issues

4. Fig 3c: It looks as if some of the *Onecut3/Slc32a1* cells may also expressed *Slc17a6* i.e., they may be *Onecut3/Glut/Gaba* triple-expressing cells.
5. Fig 3: Do all *Trh* cells express *Onecut3*?
6. Fig 4: The use of *Th* alone or *Ddc* alone is not sufficient to determine a dopaminergic identity, because both enzymes are expressed more broadly than just dopaminergic neurons.
7. Fig 5 and S6: The claim that overexpression of *Onecut3* "significantly reduces proliferation" is not well supported by the data presented. The cells could undergo apoptosis as well/instead. Cell counts and Phosphor-H3 staining would support this claim better.
8. Fig 5c: The "55,487" genes analysed, with 911 DE, must include many non-protein coding genes. Could they separate the analysis into protein coding and non-protein coding genes, and also present that data as DEup and DEdown (volcano plot or similar). In addition, a list of the 911 DE should be included as a supplementary table. Also, the statement of "GO classification detected 911 of 55,487 genes as differentially regulated..." does not make sense. Gene ontology classification refers to the molecular and biological properties of the genes being DE, not whether they are DE or not.

9. The connection between *Onecut3* and the proposed Nav2-TRIO-RhoA pathway feels somewhat taken out of thin air. Regarding candidate gene selection they “put particular emphasis on those implicated in neurite outgrowth and/or pathfinding”, but based on what criteria? GO ontology terms, literature searches? And the pathway is loosely based upon a mix of data from *C. elegans* and *Drosophila* without much support in the mammalian literature, although this part of the study is dealing with mouse cells. At the bottom of p6 they refer to that the pathway is also based upon protein-protein interaction data from ref [3]. But ref [3] is their previous scRNA-seq study and has no protein-protein interaction data.

10. ITX3 specificity? What about Crispr/Cas9 or siRNA knock-out/down of Nav2, Trio and RhoA? What about RhoA inhibitors?

11. Fig 7: *Onecut3* is not haploinsufficient in humans. What is the evidence for that reduction of *Onecut3* expression, but not loss thereof, would result in a *Onecut3*-specific phenotype. Can they repeat the siRNA experiment with other *Onecut3*-targeting siRNAs?

12. Fig 8d-d1: Why was not *unc-53* tested for chemotaxis?

13. The manuscript is well-organised and well-written. However, some of the terms and concepts used are a probably a bit unfamiliar to the developmental biologist. For instance, at the end of the Introduction they state that “...*Onecut3* recruits Nav2...”. “Recruits” is more commonly used to describe protein-protein interactions, and in this case, it would be more correct to say that *Onecut3* regulates Nav2. Also, on p5, the ‘cascade diversification hypothesis[7]’. First, ref [7] does not really outline any hypothesis to this end. Secondly, the fact that TFs act in cascades goes back to studies in many systems during the last 3 decades and is not much of a hypothesis. Third, if the authors are alluding to the fact that the *Ascl1* progenitors, and many other neural progenitors in many systems, generate different neural cell types at different developmental time point, this phenomenon is typically referred to as “temporal patterning” or “temporal progression” in progenitors.

14. The manuscript contains text with different colours, not sure what the meaning of this is.

15. Page 4: Please refer to specific panels in Fig. S4, not merely S4.

Reviewer #3 (Remarks to the Author):

The manuscript presents the expression of *Onecut3* in prenatal and postnatal mouse hypothalamus and reveals its potential role in regulating neuronal morphogenesis. Using a combination of techniques including whole-mount staining, fluorescent genetic labeling, single-cell transcriptomic analysis and in situ hybridization, the authors demonstrated that *Onecut3*⁺ is expressed in distinct neuronal populations, encompassing GABA/TH neurons in PeVN and glutamate/TRH neurons in LH. They further found that *Onecut3* promoted neuronal differentiation and morphogenesis via the Nav2-RhoA pathway. This study is interesting and would contribute to our understanding of *Onecut3*'s involvement in hypothalamus development.

1. Given the continuous expression of *Onecut3* in prenatal and postnatal mammalian hypothalamus, it is important to demonstrate whether genetic ablation of *Onecut3* prior to or at the onset of hypothalamic neurogenesis alters the fate specification of hypothalamic neurons, especially those in PeVN and LH. Due to the absence of evidence supporting the notion that *Onecut3*⁺ neurons possess longer dendritic processes compared to *Onecut3*⁻ neurons, it would be particularly intriguing and meaningful to investigate the role of *Onecut3* in neuronal specification rather than solely focusing on

neuronal morphogenesis.

2. More compelling evidences are required to address the role of *Onecut3* in neuronal morphogenesis *in vivo*. Moreover, is it possible to three-dimensionally reconstruct the morphology of *Onecut3*⁺ neurons (Figure 7b) and perform quantification? How does the authors distinguish *Onecut3*⁺ from *Onecut3*⁻ neurons in the LH with siRNA injection?

3. It would be advantageous to depict the spatial distribution of *Onecut3*⁺ neurons throughout the embryonic and postnatal hypothalamus along the rostrocaudal axis.

4. While single-cell RNA-seq data suggests that there are only two distinct populations of *Onecut3*⁺ neurons, the images in Figures 3b-3c show a sizable population of *Onecut3*⁺ neurons that are negative for *Trh* and *Th* in AHN. What is the identity of these neurons? The diversity of *Onecut3*⁺ neurons may be underestimated in the study.

5. Several magnified images are not precisely consistent with the dashed box, warranting attention to ensure accurate correspondence.

6. Some citations included in the manuscript do not align with the corresponding reference.

Univ. Prof. Dr. Tibor Harkany
Head of Department
Medical University of Vienna
**Department of Molecular Neurosci-
ences, Center for Brain Research,**
Spitalgasse 4, 1090 Vienna, Austria
T: 43 (0)1 40160 34050
E-Mail: tibor.harkany@meduniwien.ac.at

Vienna, 28-Jun-2024

RE: NCOMMS-22-47020B, point-by-point responses to the Expert Referees

We thank the three expert Reviewers for their constructive, supportive, and insightful comments on our original manuscript. We have made every attempt to experimentally address their specific queries. While, admittedly, some of the experiments were challenging and might not have returned the results what we all had hoped for, they showcase our readiness, commitment, and desire both to comply with the Reviewers' requests and to strengthen our manuscript as much as possible.

Therefore, we are cautiously optimistic that you can recommend the present version of our report for publication. We thank you for your time and energy to evaluate our submission. On behalf of my colleagues, yours sincerely:

Tibor Harkany

Point-by-point responses to Reviewer #1:

Q1: 'In the title of the article, why did the authors choose to mention "dopamine and glutamate neurons in the hypothalamus" and not dopamine and TRH neurons or GABA/tyrosine hydroxylase (TH) and glutamate/TRH to be more accurate?'

Thank you for highlighting this point. We have changed the title to both be more accurate and also better identify the major conceptual thrust of our study.

Q2: 'In contrast to TRH neurons that appear to express exclusively vesicular glutamate transporter 2, couldn't TH neurons, which are tagged by the authors as being GABAergic, also be glutamatergic? Figure 2c appears to strongly suggest this.'

The heterogeneity of dopamine neurons in the hypothalamus (with their 10 molecularly segregated groups; Romanov *et al.*, Nature (2020)) allows for both glutamate and GABA signatures. However, dopamine neurons of the **periventricular hypothalamic nucleus** are exclusively GABAergic. The overlap that the UMAP visualization suggested, is essentially non-existent (**13 out of 468 neurons with generally low transcript levels**) when addressing this at the single-cell level.

Q3: 'Did the authors assess for potential sexual dimorphisms? For example, it is well-known that, in the anteroventral periventricular region of the preoptic region, the TH neuronal population is sexually dimorphic. Could this sexual dimorphism contribute to the loss of *Onecut3* expressing cells during postnatal development (e.g., Figure 1j)?'

We have done this at two levels: 1) In our single-cell analysis, we did not find any difference in the expression patterns between neurons from male vs. female subjects (please see the relevant figures (a,a₁) on the last page of this document). For other experiments, their design excluded the specific testing of sexual dimorphism (e.g., the developing embryos were of different stages, nutrition, hormone exposure, fertilization time, number of pups). 2) While reduced levels of *Onecut3* mRNA in revised Fig. 1j can be due to neuronal loss, it can equally reflect *Onecut3* downregulation as developmental processes come to a halt in adulthood, and less *Onecut3* mRNA is required by the neurons (maintenance/remodeling phase). **Either outcome is compatible with our null hypothesis**, that *Onecut3* allows for phenotype selection and the maintenance of neuronal structure.

Q4: 'Throughout the manuscript, including page 4 and page 7, the terminology used by the authors to identify stages of postnatal development is erroneous. Indeed, P4 does not correspond to infantile mice, but neonatal mice. Similarly, P35 C57/Blk6 mice are far from being adult mice, as they actually are pre-pubertal. Postnatal development is usually described as follows: P0-P7 the neonatal period, P8-P20 the infantile period (when minipuberty occurs), P21-P35 the juvenile period, followed by the peripubertal period (puberty usually occurs around P45). Adulthood is only reached at P60.'

Thank you for this input. **We have changed the terminology throughout.** However, please note that our Steffens *et al.* (EJN, 2022) paper was massively criticized for using the above developmental scaling protocol, with the Referees and Editor jointly forcing us to use 'infant mice' already prior to P7.

Q5: 'Figure 7 is not very convincing (both a and b panels). Could the authors show the mCherry expression/immunolabelling? Could the authors provide fluorescent images with a wider field of view to see both targeted and non-targeted areas by the siRNA?'

We have **replaced the images** with fluorescent ones (**Fig. 8a**). Wide-field views would unfortunately not be helpful here because the images we show at 63x primary magnification focus on fine cellular structures. None of these details could be resolved on overview/survey images regardless of the limited space offered by the figure itself or as a separate ED Figure. Instead, we have carefully contrasted the images in **Fig. 8c**, which are grey-scaled conversions of *post-hoc* histochemistry for mCherry to find visually-pleasing yet reproducible illustrations.

Q6: 'In the *Onecut3* knockdown experiments, was *Nav2* expression downregulated, as shown for *C. Elegans* Figure 8b?'

We **have repeated the siRNA injections**, and indeed found **significant reduced *Nav2* expression also in mouse (Fig. 8a)**.

Q7: 'Figure 8. Not being familiar with *C. Elegans* neuroanatomy, this reviewer has difficult times to understand what was measured, e.g., dendritic length. Could the authors provide a schematic illustrating what they actually measured? In the results section they speak of dendritic length while in the discussion/legend they mention defasciculation...? Where are the cell bodies of the neurons? In the WT panel, one can see bilateral symmetric projections/dendrites, while in the *ceh-48* panel projections/dendrites appear to be unilateral, is this real or proper to the selected image? If real, could it be due to an alteration of the expression of guidance molecules or of their receptors? Defasciculation could be also be a result of the alteration of the chemotrophic pathways (e.g. *Robo3* or *Ephb2* orthologs).

We apologize for the confusion regarding the original set of *C. elegans* data. 1) We have **generated a schematic and increased the clarity of labeling** in the images to help conveying our message (**ED Fig. 11a**). 2) Bilateral symmetry is difficult to quantify because of the orientation of the dendrites in the images that depend on the rotation of the worm's body at mounting. While we felt that some alterations to **symmetry might indeed occur**, this was more of a subjective impression and not a line of analysis we pursued exactly because of the unsurmountable technical difficulties. 3) Indeed, downstream targets other than those we found in our sequencing study in Neuro-2A cells (see also the revised and expanded volcano plot in **ED Fig. 9**, and the protein interaction pathway in **revised Fig. 6a**) could contribute to the defasciculation phenotype. Please note that in *C. elegans* it is **the dendrites of the sensory neurons that protrude towards the mouth** in a conjoined manner that is reminiscent of axonal fasciculation in mammals. We at no point intended to allude to Nav2 being the only regulator of the phenotype observed, particularly since neuronal differentiation is a complex multifactorial process. Therefore, **we have carefully edited the manuscript** to avoid overinterpretation of the experimental results.

Point-by-point responses to Reviewer #2:

Q1: 'Onecut3 loss-of-function analysis in the mouse is based upon siRNA, which is notorious for giving only partial knock-down and for off-target effects. Onecut3 mouse mutant analysis would have been much better. If they do not have access to Onecut3 mouse mutants, analysis of additional siRNAs and/or transient Crispr/CAS9 knockout experiments would greatly strengthen the manuscript. Furthermore, are there neuronal cell lines expressing Onecut3 where its function could be probed by knock-down/knock-out?'

We appreciated your request and attention to potential experimental caveats. We have taken essentially all steps you have proposed to address this issue: 1) We have designed guide RNAs and targeting vectors to manipulate *Onecut3* in mCherry⁺ primary hypothalamic neurons using a **CRISPR/Cas9 system**. Briefly, guide RNAs were cloned into a modified lentiCRISPRv2 vector (Addgene, #52961) that additionally expressed turboGFP through an SFFV promoter. Lentiviral particles were produced in HEK293T cells and validated in Neuro-2A cells by assessing GFP expression. Most unfortunately, we were unable to achieve reproducible transfection levels in primary neurons, mostly because of low GFP expression efficiency, even when incubating them with LentiBOOST, and/or polybrene/protamine sulfate overnight. Therefore, **data were not inserted in the manuscript but are shown at the last page of this letter (panel 'b')**. 2) Live-cell imaging of fluorescently tagged neurons significantly affected their cytoskeletal dynamics, and even compromised the overall survival of the cultures (with most neurons killed by phototoxicity within 24 h (**panel 'b₁'**)). We have used an IncuCyte SX5 instrument to do these experiments because it allows for the simultaneous detection of neurite length, and be compatible with our original data and figures. We are aware that **cytotoxicity** can be minimized in, e.g., spinning-disc systems (e.g. Andor Revolution). Whilst we have made preliminary experiments on those, the throughput is inconceivably low to complete a study with the many variables, conditions, and time points as shown here. Therefore, **we are not confident to include data** involving the use of CRISPR/Cas9. 3) We have recovered (from cryopreserved stock, with a logistical effort involving institutes in 3 countries as intermediary points of work) *Onecut3* null mice. When performing qPCR for *Nav2* in hypothalami at E14.5, we did not find any change in *Nav2* mRNA expression (**data are as panels 'c-c₂' in the figure on the last page**). Similarly, we could not morphometrically resolve major changes to neuronal structure in the periventricular area. We attributed these shortcomings to the fact that the constitutive deletion of genes notoriously leads to compensation, especially during development. This is particularly true for the *Onecut* family, as single mutants rarely demonstrate phenotypic changes as demonstrated, e.g., in mammalian pancreatic cells (Vanhorenbeeck, V. *et al.* Role of the Onecut transcription factors in pancreas morphogenesis and in pancreatic and enteric endocrine differentiation. *Dev. Biol.* **305**, 685–

694 (2007)), as well as in single CUT-mutant *C. elegans* strains. In contrast, compound mutants exhibit striking disturbances (Leyva-Díaz, E. & Hobert, O. Robust regulatory architecture of pan-neuronal gene expression. *Curr. Biol. CB* **32**, 1715-1727.e8 (2022)). To overcome these problems, we have repeated the **siRNA experiment with a pool of 4 RNAi probes**, quantified *Nav2* mRNA levels in *Onecut3⁺* neurons, and found them significantly reduced (**revised Fig. 8b**). Therefore, we have exhausted the experimental repertoire available to us, and also improved the original dataset considerably.

Q2a: 'The manuscript does not articulate what specific neuronal property/properties *Onecut3* controls? All neurons extend dendrites and axons, so this neuronal cellular polarity is not a feature specific to *Onecut3* neurons. Do *Onecut3* neurons project axons and/or dendrites to a common target(s)? Do they have unusually long/short dendrites/axons? Do they connect to a common type of pre/post-synaptic neuron?'

We have recently published the brain-wide connectivity of both periventricular TH⁺ and lateral hypothalamic TRH⁺ neuron populations in the adult brain (Zupančič, M. *et al.* Brain-wide mapping of efferent projections of glutamatergic (*Onecut3⁺*) neurons in the lateral mouse hypothalamus. *Acta Physiol.* 2023, e13973; Korczynska, S. *et al.* A hypothalamic dopamine locus for psychostimulant-induced hyperlocomotion in mice. *Nat. Commun.* 13, 5944 (2022)). In these studies, we found that ***Onecut3⁺* neurons target other midline regions rich in *Onecut3⁺* neurons** (e.g., lateral septum and lateral habenula). Moreover, they **converge on output targets with abundant dopamine/TRH receptor expression**. Thus, there is reason to believe that GABAergic vs. glutamatergic *Onecut3⁺* neurons could cross-modulate shared postsynaptic targets. Thereby, we validate two of your inferences: 1) *Onecut3⁺* neurons have long axons. Indeed, this is already evident at mid-gestation, both in the telencephalon and the mesencephalon/spinal nerves from E14.5. 2) There is a need for the molecular machinery to drive long-range axon development in *Onecut3⁺* neurons. Undoubtedly, these neurons also use classical chemotropic cues for development. We assume, based on our single-cell RNA-seq data, that e.g., ErbB2, can be regulated by *Onecut3*, which is a pathway mandatory to retain cytoskeletal instability sufficient for, e.g., correcting guidance decisions. However, our conclusion is that *Onecut3* controls a cell autonomous signaling cascade that allows for neurite extension, rather than navigation, in the developing nervous system.

Q2b: 'Similarly, what is the common evolutionarily conserved role of *Onecut3/ceh-48*? They state that both genes control "neuronal morphogenesis", but this is a rather broad phenotype.'

Onecut/Ceh-48 has been shown to induce neuron-like identity when overexpressed in glia cells in *C. elegans* (Leyva-Díaz, E. & Hobert, O. Robust regulatory architecture of pan-neuronal gene expression. *Curr. Biol. CB* **32**, 1715-1727.e8 (2022)). Thus, *Onecut/Ceh-48* is **sufficient to trigger fate decision biased towards neurogenesis**. We found a similar choice bias when overexpressing OC3 in a mammalian glioblastoma cell line, inducing neuron-like morphology (including the induction of MAP2 and TUJ1, and reduction of GFAP; **ED Fig. 7b**). These data unite in the proposal that ***Onecut3* promotes the transcription of pro-neuronal genes**, irrespective of either the model organism or the cell type. Since Rho-GTPases are ubiquitous in developing neurons, *Onecut3* – particularly when overexpressed – can recruit a signaling pathway to drive neurogenesis.

Q3: 'Fig 3c: It looks as if some of the *Onecut3/Slc32a1* cells may also expressed *Slc17a6* i.e., they may be *Onecut3/Glut/Gaba* triple-expressing cells.'

This is correct. Thirteen out of 468 cells expressed both *Slc31a1* and *Slc17a6* (**Fig. 2c**), which we attribute to the remnant of *Onecut1/2* regulation of neuronal fate decision rather than a specific effect of *Onecut3*.

Q4: 'Fig 3: Do all Trh cells express *Onecut3*?'

We find **~29% of all *Trh*⁺ neurons to express *Onecut3***. This observation points to the inherent heterogeneity of TRH neurons. However, this topic was outside the scope of the present study, and focused on *Onecut3* instead.

Q5: 'Fig 4: The use of Th alone or Ddc alone is not sufficient to determine a dopaminergic identity, because both enzymes are expressed more broadly than just dopaminergic neurons.'

Thank you for asking us to improve this definition. Indeed, we have specified the molecular constituents that define *bona fide* dopaminergic identity of *Onecut3*⁺ neurons in the periventricular nucleus recently (Korchynska, S. *et al.* A hypothalamic dopamine locus for psychostimulant-induced hyperlocomotion in mice. *Nat. Commun.* 13, 5944 (2022)). We have **updated the text to clarify and reflect this molecular definition**.

Q6: 'Fig 5 and S6: The claim that overexpression of *Onecut3* "significantly reduces proliferation" is not well supported by the data presented. The cells could undergo apoptosis as well/instead. Cell counts and Phosphor-H3 staining would support this claim better.'

We have significantly strengthened this point experimentally. As a recap though, our live-cell imaging experiments (**Fig. 5d₁**) showed the 'cell cluster area', which is a measure of the number of cell bodies in any given field of view. At baseline, this value was equivalent across all conditions, pointing to no *a priori* experimental bias. At no point from transfection on (regardless of the type of the plasmids) had this variable dropped, meaning there was continuous cell proliferation across the samples. The differences (positive deviation over time) suggested different rates of proliferation. Images in **Figs. 5c,d** also failed to show any damaged or dead cells. Thus, the experiment was technically sound. Nevertheless, and to concur with your request, we have repeated the overexpression experiments in **Neuro-2a** cells and labelled those for the mitotic marker pHH3. Significantly reduced numbers of **pHH3⁺ cells** were found, confirming reduced cell proliferation. *Onecut3* co-labeling was rarely, if ever, found (**Fig. 5a, a₁**). Even more importantly, immunocytochemistry for **cleaved caspase-3**, a direct mark of apoptosis, ruled out excessive cell death due to *Onecut3* overexpression (**Fig. 5b**). Thus, our data **point to the control of the rate of cell proliferation**.

Q7: 'Fig 5c: The "55,487" genes analysed, with 911 DE, must include many non-protein coding genes. Could they separate the analysis into protein coding and non-protein coding genes, and also present that data as DEup and DEdown (volcano plot or similar). In addition, a list of the 911 DE should be included as a supplementary table. Also, the statement of "GO classification detected 911 of 55,487 genes as differentially regulated..." does not make sense. Gene ontology classification refers to the molecular and biological properties of the genes being DE, not whether they are DE or not.'

Indeed, our wording was least helpful. We have **revised this part of the manuscript** with care. Moreover, we have added a full volcano plot (**revised ED Fig. 9**). The **complete list of genes** (both up- and down-regulated) **was attached** as an Excel spreadsheet. The **GO classification** was also clarified.

Q8: 'The connection between *Onecut3* and the proposed Nav2-TRIO-RhoA pathway feels somewhat taken out of thin air. Regarding candidate gene selection they "put particular emphasis on those implicated in neurite outgrowth and/or pathfinding", but based on what criteria? GO ontology terms, literature searches? And the pathway is loosely based upon a mix of data from *C. elegans* and *Drosophila* without much support in the mammalian literature, although this part of the study is dealing with mouse cells. At the bottom of p6 they refer to that the pathway is also based upon protein-protein interaction data from ref [3]. But ref [3] is their previous scRNA-seq study and has no protein-protein interaction data.'

Candidate gene selection centered on the observation that overexpression of *Onecut3* induced neurite outgrowth (cytoskeletal modification). Accordingly, we have performed a **protein-interaction pathway reconstruction** using **Knit** and **Omnipath with the mouse as reference organism**. We have **added the entire reconstructed network** in revised **Fig. 6a**, differentiated up- and down-regulated components, and defined their relationship to *Onecut3*. The **corresponding text was also revised**, including a note that the arrows do not indicate directionality *per se* but show interactions between protein pairs. We have also **changed the reference for the protein interaction pathway algorithm**.

Q9: 'ITX3 specificity? What about Crispr/Cas9 or siRNA knock-out/down of Nav2, Trio and RhoA? What about RhoA inhibitors?'

ITX3 is considered a specific and non-toxic inhibitor of TRIO (Bouquier, N. et al. A cell active chemical GEF inhibitor selectively targets the Trio/RhoG/Rac1 signaling pathway. *Chem. Biol.* 16, 6 (2009)). Yet, and additionally, we aimed at verifying the role of Nav2 in neurite extension using Neuro-2a cells manipulated by CRISPR/Cas9 editing (alike for Q1). After infection, cells expressing the empty vector had significantly delayed proliferation as compared to untransfected cells (see **Figures d,d₁ overleaf vs. ED Fig. 10a₁**). Co-transfection with *Onecut3* did neither reduce proliferation further nor induced neurite outgrowth (**Figure d₂ overleaf**). **We attribute the lack of response to cellular metabolic constraints, because these cells needed to be transfected with plasmids containing strong promoters to express multiple proteins** (e.g. tGFP and Cas9). As an alternative, we have exposed *Onecut3*-transfected Neuro-2A cells to Y-27632, which inhibits Rho kinase (ROCK). Unexpectedly, yet in line with literature (Park, S. Y., An, J. M., Seo, J. T. & Seo, S. R. Y-27632 Induces Neurite Outgrowth by Activating the NOX1-Mediated AKT and PAK1 Phosphorylation Cascades in PC12 Cells. *Int. J. Mol. Sci.* 21, 7679 (2020)), Y-27632 did not reduce neurite outgrowth. **Instead, Y-28632 vastly increased neurite length in transfected cells (Fig. S10a,a₁), with an effect additive to that of Onecut3. These data have been presented and also discussed** in the revised paper.

Q10: 'Fig 7: Onecut3 is not haploinsufficient in humans. What is the evidence for that reduction of Onecut3 expression, but not loss thereof, would result in a Onecut3-specific phenotype. Can they repeat the siRNA experiment with other Onecut3-targetting siRNAs?'

Previously (Romanov, R. A. et al. Molecular design of hypothalamus development. *Nature* 1–7 (2020) doi:10.1038/s41586-020-2266-0.), we analyzed genetic variations of *Onecut3* in GWAS data and found not a single *Onecut3* SNP associated with any disease. This indicates a strong purifying selection drive with **mutations in Onecut3 directly and adversely impacting survival**. Indeed, the functional importance of *Onecut3* is stressed by its placement on a “run of homozygosity” cold-spot region, a genomic sequence considered capable of avoiding purely lethal or cryptic mutation-critical functions (Pemberton, T. J. et al. Genomic Patterns of Homozygosity in Worldwide Human Populations. *Am. J. Hum. Genet.* 91, 275–292 (2012)). Therefore, we remain firm in our view that **even partial siRNA-mediated knockdown of Onecut3 would reveal a phenotype stronger than full genetic loss**, especially considering compensatory mechanisms by other *Onecuts* in both mammalian and *C. elegans* knock-out models (Vanhorenbeeck, V. et al. Role of the *Onecut* transcription factors in pancreas morphogenesis and in pancreatic and enteric endocrine differentiation. *Dev. Biol.* 305, 685–694 (2007); Leyva-Díaz, E. & Hobert, O. Robust regulatory architecture of pan-neuronal gene expression. *Curr. Biol. CB* 32, 1715-1727.e8 (2022)). Here, we have repeated the siRNA experiment, as per your suggestion, with a **SMARTpool of 3-4 siRNAs to maximize the specific targeting of Onecut3. This led to a significant reduction in both Onecut3 and Nav2 in the neuron populations sampled (Fig. 8a).**

Q11: 'Fig 8d-d1: Why was not *unc-53* tested for chemotaxis?'

Chemotaxis experiments have been repeated, including *unc-53*. Both ***unc-53* and *ceh-48* loss-of-function phenocopied one other**, with a significant reduction in responses (**revised Fig. 9d**).

Q12: 'The manuscript is well-organised and well-written. However, some of the terms and concepts used are a probably a bit unfamiliar to the developmental biologist. For instance, at the end of the Introduction they state that "...Onecut3 recruits Nav2...". "Recruits" is more commonly used to describe protein-protein interactions, and in this case, it would be more correct to say that Onecut3 regulates Nav2. Also, on p5, the 'cascade diversification hypothesis[7]'. First, ref [7] does not really outline any hypothesis to this end. Secondly, the fact that TFs act in cascades goes back to studies in many systems during the last 3 decades and is not much of a hypothesis. Third, if the authors are alluding to the fact that the *Ascl1* progenitors, and many other neural progenitors in many systems, generate different neural cell types at different developmental time point, this phenomenon is typically referred to as "temporal patterning" or "temporal progression" in progenitors.'

We apologize for the inaccurate use of terms. **We have corrected these inconsistencies.**

Q13: 'The manuscript contains text with different colours, not sure what the meaning of this is.'

These colors were used to indicate changes requested by the handling editor on an earlier version of this submission. Here, **we have highlighted all changes made during revision in blue** for ease of assessing our additions.

Q14: 'Page 4: Please refer to specific panels in Fig. S4, not merely S4.'

The **text has been edited** as requested.

Point-by-point responses to Reviewer #3:

Q1: 'Given the continuous expression of *Onecut3* in prenatal and postnatal mammalian hypothalamus, it is important to demonstrate whether genetic ablation of *Onecut3* prior to or at the onset of hypothalamic neurogenesis alters the fate specification of hypothalamic neurons, especially those in PeVN and LH.'

To answer your query, we have obtained *Onecut3* heterozygous and *Onecut3* knockout hypothalami from E14.5 embryos (after resuscitating the line from cryopreserved stock). Quantitative PCR showed no difference in *Onecut1*, *Onecut2*, and *Nav2* mRNA relative to controls (see Figure panels c-c₂ overleaf), nor did we find any apparent change to dopaminergic neuron morphology. This was not entirely unexpected, as *Onecuts* share common downstream pathways (Romanov, R. A. *et al.* Molecular design of hypothalamus development. *Nature* 1–7 (2020) doi:10.1038/s41586-020-2266-0.), and thus are prone to compensate for each other during developmental processes. Accordingly, single *Onecut* mutants do not show any major developmental phenotypes during either mammalian or *C. elegans* cell development. In contrast, compound mutants present robust phenotypes (Vanhorenbeeck, V. *et al.* Role of the *Onecut* transcription factors in pancreas morphogenesis and in pancreatic and enteric endocrine differentiation. *Dev. Biol.* **305**, 685–694 (2007); Leyva-Díaz, E. & Hobert, O. Robust regulatory architecture of pan-neuronal gene expression. *Curr. Biol. CB* **32**, 1715-1727.e8 (2022)). **To minimize the likelihood of compensation mechanisms, we have repeated the siRNA experiments and now show that *Nav2* mRNA is significantly reduced when *Onecut3* is knocked down (Fig. 8b).**

Q2: 'Due to the absence of evidence supporting the notion that *Onecut3*⁺ neurons possess longer dendritic processes compared to *Onecut3*⁻ neurons, it would be particularly intriguing

and meaningful to investigate the role of *Onecut3* in neuronal specification rather than solely focusing on neuronal morphogenesis.'

To address this issue, **we transfected cortical and hypothalamic neural stem cells obtained from E14.5 mouse pups with *Onecut3* (revised ED Fig. 8)**. We found that transfected cultures had a significant reduction in cluster size, indicating a decrease in proliferation. Indeed, we found **a lack of the mitotic marker pHH3 in *Onecut3*⁺ cells residing within the clusters**. Finally, we found a strong **reduction of SOX2** (controlling neural proliferation), indicating that cells cease to divide and be primed for differentiation when *Onecut3* is present.

Q3: 'More compelling evidences are required to address the role of *Onecut3* in neuronal morphogenesis in vivo. Moreover, is it possible to three-dimensionally reconstruct the morphology of *Onecut3*⁺ neurons (Figure 7b) and perform quantification? How does the authors distinguish *Onecut3*⁺ from *Onecut3*⁻ neurons in the LH with siRNA injection?'

Due to the density of *Onecut3*⁺ neurons in the lateral hypothalamus, especially during *in utero* development where the amount of astrocytes is limited and the brain is still expanding, it is challenging to reconstruct the morphology of single *Onecut3*⁺ neurons. Therefore, we opted to measure the area coverage of mCherry⁺ fibers after siRNA vs. control injections locally. Since we are aware of the limitations of siRNA experiments, including its limited half-life, variable knockdown efficacy, and the *post-hoc* identification of transfected cells in mammals, as well as the difficulties associated with *Onecut* knockout models (see also comments to Reviewer 2), **we have favored *C. elegans* as an experimental subject to address this query (revised Fig. 9)**. We found equivalent morphological phenotypes, that is, **reduced neuronal complexity, when deleting either *Onecut3* (*ceh-48*) or *Nav2* (*unc-53*)**. We have performed additional chemotaxis experiments to show that these neuronal changes have significant behavioral impact (Fig. 9d₁).

Q4: 'It would be advantageous to depict the spatial distribution of *Onecut3*⁺ neurons throughout the embryonic and postnatal hypothalamus along the rostrocaudal axis.'

We have presented the **mRNA distribution patterns for *Onecut3*, *Trh* and *Th* from preoptic → anterior → tuberal regions for E14.5 and P3 in revised Fig. 3a-d₂**. We additionally provided the immunostainings for ONECUT3 and TH in ED Fig. 4a-b₂. For the extensive mapping in the adult, we direct you to our recent paper that shows the brain-wide connectivity of *Onecut3*⁺ neurons (Zupančič, M. *et al.* Brain-wide mapping of efferent projections of glutamatergic (*Onecut3*⁺) neurons in the lateral mouse hypothalamus. *Acta Physiol.* 2023, e13973)).

Q5: 'While single-cell RNA-seq data suggests that there are only two distinct populations of *Onecut3*⁺ neurons, the images in Figures 3b-3c show a sizable population of *Onecut3*⁺ neurons that are negative for *Trh* and *Th* in AHN. What is the identity of these neurons? The diversity of *Onecut3*⁺ neurons may be underestimated in the study.'

Indeed, there is a significant population of *Slc32a1/Onecut3*⁺ neurons stretching along the rostrocaudal axis of the hypothalamus. However, due to **the lack of particular cellular identifiers, such as neurotransmitters and neuropeptides, these cannot be unequivocally classified as distinct neuronal populations**. We posit these cells to be a cohort of intermediate neurons, not yet committed to a terminal identity. This concurs with our observation that approximately 50% of GABA cells remain in transitory phases for extended periods during mid-gestation in the mouse hypothalamus (Romanov, R. A. *et al.* Molecular design of hypothalamus development. *Nature* 1–7 (2020) doi:10.1038/s41586-020-2266-0.), and also coincides with the recently identified hypothalamic "ghost cells" which only become phenotypically distinct under certain conditions (Leon, S. *et al.* Single cell tracing of Pomc neurons reveals recruitment of 'Ghost' subtypes with atypical identity in a mouse model of obesity. *Nat Commun.* 15, 3443 (2024)).

Q6: 'Several magnified images are not precisely consistent with the dashed box, warranting attention to ensure accurate correspondence.'

Due to the **labeling inside the boxes**, the box sizes have been increased to prevent cluttering. We have **revisited these** and were as precise as possible.

Q7: 'Some citations included in the manuscript do not align with the corresponding reference.'

We have double-checked these, and **corrected as necessary**.

Figure for review. Additional data collected to address the Reviewers' queries. **(a,a₁)** Male and female *Onecut3* distribution plots showed similar patterns across the cell populations tested, suggesting no apparent sex differences. **(b)** Primary hypothalamic cultures from E14.5 *Onecut3*-mCherry⁺ pups (*arrowheads*) were transfected with a CRISPR/Cas9 plasmid containing single guide RNAs for *Onecut3*, and processed after 6 days in vitro (DIV). Note the low transfection efficiency (GFP, *open arrowheads* vs. *number of surrounding nuclei*). **(b₁)** Incucyte-based live-cell imaging (every 4h) for phase contrast and GFP negatively impacted neuronal survival after 24h (*open arrowheads*). **(c-c₂)** Quantitative PCR from heterozygous and knockout *Onecut3* E14.5 hypothalami did not show any differences in *Onecut1*, *Onecut2* and *Nav2* mRNA levels between groups ($n = 7-8$). **(d-d₂)** Neuro-2A cells overexpressing a CRISPR/Cas9 plasmid targeting *Nav2* showed significantly delayed proliferation (CD8, control; SgX, single guide RNA), and did not respond to *Onecut3* overexpression (**d₂**).

REVIEWER COMMENTS

Reviewer #1 (Remarks to the Author):

The authors satisfactorily answered the Reviewers' comments.

Reviewer #2 (Remarks to the Author):

The re-submitted manuscript has addressed several of the issues that I raised regarding the original submission, and this has greatly improved the manuscript. However, several issues have not been addressed. In addition, the new findings of a lack of phenotype in *Onecut3* mouse knockouts is quite concerning.

For clarity, I will refer to my original numbered points.

Major issues

1. I requested additional experiments regarding the *Onecut3* loss-of-function analysis, suggesting analysis of additional siRNAs and/or transient Crispr/CAS9 knockout experiments and/or *Onecut3* mouse mutant analysis. The authors have pursued all three avenues. They successfully show that a mix of 4 RNAi probes targeting *Onecut3* also affects *Nav2* mRNA levels. However, regarding knockout of *Onecut3* in cell lines/primary cells they were unable to successfully complete the study due to technical issue. More concerningly, they did analyse *Onecut3* mutant mice, but found no effect on *Nav2* mRNA levels. They argue for developmental compensatory effects. However, their analysis reveals no change in *Onecut1* or -2 mRNA levels in the *Onecut3* knockout mice. This would imply developmental compensatory effects involving genes other than *Onecut* family members, and that their RNAi knock-down of *Onecut3* has effects because it is being conducted in adult mice, avoiding developmental compensatory effects. This is possible, but seeing conflicting results between siRNA (effects) and genetic knockout (no effects) raises serious concerns.

2. OK.

3. "Fig 7: Does *Onecut3* knock-down result in changes in neurotransmitter expression?" NOT ANSWERED.

Minor issues

4. OK.

5. OK.

6. OK.

7. I requested further support for the claim that overexpression of *Onecut3* "significantly reduces proliferation". They have now conducted cell counts, apoptosis staining and *Onecut3*/Phosphor-H3 staining. But the only effect that appears significant is the fact that *Onecut3* never overlaps with pHH3 (Fig 5a1). Please clarify the statistical analysis of cell numbers, pHH3 numbers and Cleaved-Caspase-3 numbers. If none of these numbers are significantly affected by *Onecut3* overexpression the claims of *Onecut3* reducing proliferation must be toned down.

8. OK.

9. OK.

10. My original concerns were: "ITX3 specificity? What about Crispr/Cas9 or siRNA knock-out/down of Nav2, Trio and RhoA? What about RhoA inhibitors?" They now report on attempts to test knock-down of Nav2 in Neuro-2a cell culture but observed no effect. But they also observed no effect of Onecut3 overexpression in these cells and concluded that there were technical issues. Next, they used a Rho kinase inhibitor, finding that it significantly increased neurite length. But there is no statistical analysis backing up this claim, only the graphs (Fig S10a, a1). Please clarify the statistics for this claim.

11. My concerns: "Fig 7: Onecut3 is not haploinsufficient in humans. What is the evidence for that reduction of Onecut3 expression, but not loss thereof, would result in a Onecut3-specific phenotype. Can they repeat the siRNA experiment with other Onecut3-targetting siRNAs?" They claim that "...even partial siRNA-mediated knockdown of Onecut3 would reveal a phenotype stronger than full genetic loss..." due to "compensatory mechanisms". This makes no sense. However, on the plus side and as noted above, my point 1, they have tested other RNAi probes and again observed effects of Onecut3 knock-down.

12. OK.

13. OK.

14. OK.

15. OK.

Reviewer #3 (Remarks to the Author):

All of the questions and concerns have been addressed by the authors.

Univ. Prof. Dr. Tibor Harkany
Head of Department
Medical University of Vienna
**Department of Molecular Neurosci-
ences, Center for Brain Research,**
Spitalgasse 4, 1090 Vienna, Austria
T: 43 (0)1 40160 34050
E-Mail: tibor.harkany@meduniwien.ac.at

Vienna, 06-Aug-2024

RE: NCOMMS-22-47020C, point-by-point responses to the Expert Referees

We thank the three expert Reviewers for their time and effort in evaluating our manuscript.

We were most glad to learn that **Reviewers 1 and 3 accepted our manuscript 'as is', and recommended publication.**

Likewise, we thank **Reviewer 2** for asking additional clarifications and the discussion of additional mechanistic details. Please find our replies to their specific queries below.

Overall, we are optimistic that you can recommend our study for publication in its present form. Thank you for your time to evaluate our report.

On behalf of my colleagues,
Yours sincerely:

Tibor Harkany

Point-by-point responses to Reviewer #1:

SUMMARY 1: *'The authors satisfactorily answered the Reviewers' comments.'*

Thank you for approving our manuscript for publication.

Point-by-point responses to Reviewer #2:

SUMMARY 2: *'The re-submitted manuscript has addressed several of the issues that I raised regarding the original submission, and this has greatly improved the manuscript. However, several issues have not been addressed. In addition, the new findings of a lack of phenotype in Onecut3 mouse knockouts is quite concerning. For clarity, I will refer to my original numbered points.'*

Thank you for recognizing the quality and quantity of our revisions, and considering our manuscript as *'greatly improved'*. Please find our point-by-point responses to your remaining concerns as follows.

Q1: *'Major issues: I requested additional experiments regarding the Onecut3 loss-of-function analysis, suggesting analysis of additional siRNAs and/or transient Crispr/CAS9 knockout ex-*

periments and/or *Onecut3* mouse mutant analysis. The authors have pursued all three avenues. They successfully show that a mix of 4 RNAi probes targeting *Onecut3* also affects *Nav2* mRNA levels. However, regarding knockout of *Onecut3* in cell lines/primary cells they were unable to successfully complete the study due to technical issue. More concerningly, they did analyse *Onecut3* mutant mice, but found no effect on *Nav2* mRNA levels. They argue for developmental compensatory effects. However, their analysis reveals no change in *Onecut1* or *-2* mRNA levels in the *Onecut3* knockout mice. This would imply developmental compensatory effects involving genes other than *Onecut* family members, and that their RNAi knock-down of *Onecut3* has effects because it is being conducted in adult mice, avoiding developmental compensatory effects. This is possible, but seeing conflicting results between siRNA (effects) and genetic knockout (no effects) raises serious concerns.¹

We appreciated your request and attention to potential experimental and/or conceptual caveats. The expression of transcription factors that contribute to dynamic processes, such as the cell cycle, is spatiotemporally tightly controlled to regulate transcription^{1,2}. However, in less dynamic systems, including post-mitotic neurons, transcription factors regulating cell-fate are necessary only transiently to open chromatin (a likely permanent consequence of their activity) for long-term changes to occur to the transcriptional landscape¹. *Onecut3* acts on cellular maturation through chromatin remodeling of genes involved in differentiation (Figure 5e, Extended Figure 9)³. Since the binding motifs are highly homologous between all members of the *Onecut* family, pre-existing *Onecut1/2* might (partly) compensate for a functional loss of *Onecut3*^{4,5}, and do so even in a longer period of time⁶. Therefore, and even if we do not find increased amounts of *Onecut1/2* mRNA in the *Onecut3* knockout, the available pool of *Onecut1/2* proteins shall be sufficient to mitigate the lack of *Onecut3* through them only needed temporarily to modify select chromatin regions in post-mitotic neurons.

Notably, there is not a single mutation detected in *ONECUT3* in human populations (<https://www.ukbiobank.ac.uk/>)⁷, as this gene is protected against mutant accumulation via strong cross-over recombination⁸. **This means that defective *ONECUT3* is considered worse than the complete loss of *ONECUT3* as compensation by *ONECUT1/2* will not occur as defective *ONECUT3* protein remains during development.** We therefore suggest that siRNA-mediated knock-down of *Onecut3* shall be more detrimental as compensation would not be triggered instantly, which we indeed found in our analysis (Figure 8).

We apologize for not explaining in sufficient detail that *Onecut3* is a powerful chromatin remodeler, and how it controls transcription in in both proliferating cells and post-mitotic neurons. **This has been rectified in the present version of the manuscript.**

References:

1. Lu, F. & Lionnet, T. Transcription Factor Dynamics. *Cold Spring Harb. Perspect. Biol.* **13**, a040949 (2021).
2. Zaret, K. S. & Mango, S. E. Pioneer transcription factors, chromatin dynamics, and cell fate control. *Curr. Opin. Genet. Dev.* **37**, 76–81 (2016).
3. van der Raadt, J., van Gestel, S. H. C., Nadif Kasri, N. & Albers, C. A. *ONECUT* transcription factors induce neuronal characteristics and remodel chromatin accessibility. *Nucleic Acids Res.* **47**, 5587–5602 (2019).
4. Vanhorenbeeck, V., Jacquemin, P., Lemaigre, F. P. & Rousseau, G. G. OC-3, a Novel Mammalian Member of the *ONECUT* Class of Transcription Factors. *Biochem. Biophys. Res. Commun.* **292**, 848–854 (2002).
5. Leyva-Díaz, E. & Hobert, O. Robust regulatory architecture of pan-neuronal gene expression. *Curr. Biol. CB* **32**, 1715–1727.e8 (2022).
6. Garcia, D. A. *et al.* Power-law behavior of transcription factor dynamics at the single-molecule level implies a continuum affinity model. *Nucleic Acids Res.* **49**, 6605–6620 (2021).
7. Romanov, R. A. *et al.* Molecular design of hypothalamus development. *Nature* 1–7 (2020) doi:10.1038/s41586-020-2266-0.
8. Pemberton, T. J. *et al.* Genomic Patterns of Homozygosity in Worldwide Human Populations. *Am. J. Hum. Genet.* **91**, 275–292 (2012).

Q2: 'original query Q3. "Fig 7: Does *Onecut3* knock-down result in changes in neurotransmitter expression?" NOT ANSWERED.'

Thank you for pointing out the need to address this point in more detail. We have additionally quantified *Slc17a6* (VGLUT2) mRNA levels in siRNA-treated pups and found no significant change ($n = 3$ pups/condition, $P = 0.356$). Furthermore, we did not observe any change in tyrosine hydroxylase immunoreactivity in *Onecut3* null mice either.

Q3: 'original query Q7. Minor issues: I requested further support for the claim that overexpression of *Onecut3* "significantly reduces proliferation". They have now conducted cell counts, apoptosis staining and *Onecut3*/Phosphor-H3 staining. But the only effect that appears significant is the fact that *Onecut3* never overlaps with pHH3 (Fig 5a1). Please clarify the statistical analysis of cell numbers, pHH3 numbers and Cleaved-Caspase-3 numbers. If none of these numbers are significantly affected by *Onecut3* overexpression the claims of *Onecut3* reducing proliferation must be toned down.'

As shown in Figure 5a (arrow), Figure 5a₁ and the Source Data File, we do find occasional pHH3 expression in *Onecut3*⁺ cells (~6.4%), which has also been visualized by the red column part in the second normalized bar graph in Figure 5a₁. As we do find a **statistically significant reduction of pHH3 in transfected cells** ($P < 0.001$) we stated that '*Onecut3* inhibits cell proliferation, a finding that is also shown in Figures 5d₁, as well as Extended Data Figures 8a,b and 10a₁ (see also Referee query 1 and 10)'. Therefore, we are confident that *Onecut3* indeed can abrogate cell proliferation in favor of cell differentiation/maturation.

Q4: 'original query Q10. My original concerns were: "ITX3 specificity? What about *Crispr/Cas9* or siRNA knock-out/down of *Nav2*, *Trio* and *RhoA*? What about *RhoA* inhibitors?" They now report on attempts to test knock-down of *Nav2* in *Neuro-2a* cell culture but observed no effect. But they also observed no effect of *Onecut3* overexpression in these cells and concluded that there were technical issues. Next, they used a *Rho* kinase inhibitor, finding that it significantly increased neurite length. But there is no statistical analysis backing up this claim, only the graphs (Fig S10a, a1). Please clarify the statistics for this claim.'

Thank you for pointing this out. We have added the statistics to the Source Data File to demonstrate that both *Onecut3* and Y-27632 significantly increase neurite outgrowth. Asterisks have also been added to **Extended Figure 10** to indicate significance (see also to the left herein).

Q5: 'original query Q11. My concerns: "Fig 7: *Onecut3* is not haploinsufficient in humans. What is the evidence for that reduction of *Onecut3* expression, but not loss thereof, would result in a *Onecut3*-specific phenotype. Can they repeat the siRNA experiment with other *Onecut3*-targeting siRNAs?" They claim that "...even partial siRNA-mediated knockdown of *Onecut3* would reveal a phenotype stronger than full genetic loss..." due to "compensatory mechanisms". This makes no sense. However, on the plus side and as noted above, my point 1, they have tested other RNAi probes and again observed effects of *Onecut3* knock-down.'

We have addressed this concern in Q1 (above) and are grateful that the Referee appreciates our additional siRNA experiments.

Point-by-point responses to Reviewer #3:

SUMMARY 3: *'All of the questions and concerns have been addressed by the authors.'*

Thank you for appreciating the completeness of our revisions, and approving our paper as is.

REVIEWERS' COMMENTS

Reviewer #2 (Remarks to the Author):

Q1: The lack of a phenotype in the *Onecut3* knockout mice in contrast to the adult siRNA phenotypes remains a concern for me. Their reasoning of a developmental compensation by *Onecut1/2* in *Onecut3* KO vs more acute effects of adult *Onecut3* siRNA knock-down makes some sense, but as a geneticist I am always concerned when there is no KO phenotype.

Q2-5: All good.

Univ. Prof. Dr. Tibor Harkany
Head of Department
Medical University of Vienna
**Department of Molecular Neurosci-
ences, Center for Brain Research,**
Spitalgasse 4, 1090 Vienna, Austria
T: 43 (0)1 40160 34050
E-Mail: tibor.harkany@meduniwien.ac.at

Vienna, 03-Sep-2024

RE: NCOMMS-22-47020C, point-by-point response to Expert Referee 2

We thank **Reviewer 2** for accepting the additional data and explanations that we provided.

On behalf of my colleagues,
Yours sincerely:

Tibor Harkany

Point-by-point responses to Reviewer #2:

Q1: The lack of a phenotype in the *Onecut3* knockout mice in contrast to the adult siRNA phenotypes remains a concern for me. Their reasoning of a developmental compensation by *Onecut1/2* in *Onecut3* KO vs more acute effects of adult *Onecut3* siRNA knock-down makes some sense, but as a geneticist I am always concerned when there is no KO phenotype.

We appreciate your concerns and agree that loss-of-function intuitively should lead to phenotypical changes. However, genetic redundancy leading to compensation is a powerful evolutionary survival strategy as seen in for example KO models of calcium binding proteins where total loss did not lead to any phenotypical changes at all (doi.org/10.1073/pnas.060525210) through compensation by other calcium-modulating proteins. This also rings true for zebrafish research, where morpholino-based knockdown can result in stronger phenotypes as compared to full genetic knockouts ([doi.org/ 10.1016/j.gendis.2021.12.003](https://doi.org/10.1016/j.gendis.2021.12.003)). This might very well be the case for *Onecut3* as well, where other CUT factors in addition to *ONECUT1/2*, or even structurally unrelated transcription factors, could allow for the normal propagation of downstream transcriptional events.

Q2-5: All good.

Thank you.